# Learning where to learn:
# Gradient sparsity in meta and continual learning

**Johannes von Oswald\*,[1], Dominic Zhao\*,[1],**
**Seijin Kobayashi[1], Simon Schug[1], Massimo Caccia[2],**
**Nicolas Zucchet[1], João Sacramento[1]**

*Equal contribution

[1]Institute of Neuroinformatics, University of Zürich and ETH Zürich
[2]Mila, University of Montreal & ServiceNow
{voswaldj,dozhao}@ethz.ch

## Abstract

Finding neural network weights that generalize well from small datasets is difficult. A promising approach is to learn a weight initialization such that a small number of weight changes results in low generalization error. We show that this form of meta-learning can be improved by letting the learning algorithm decide which weights to change, i.e., by learning where to learn. We find that patterned sparsity emerges from this process, with the pattern of sparsity varying on a problem-by-problem basis. This selective sparsity results in better generalization and less interference in a range of few-shot and continual learning problems. Moreover, we find that sparse learning also emerges in a more expressive model where learning rates are meta-learned. Our results shed light on an ongoing debate on whether meta-learning can discover adaptable features and suggest that learning by sparse gradient descent is a powerful inductive bias for meta-learning systems.

## 1 Introduction

Meta-learning holds the promise of discovering inductive biases that improve the performance of a primary learning process. Such a set of assumptions can materialize in various elements of the learner. The well-known model-agnostic meta-learning [MAML; 11] algorithm aims to learn a neural network initialization that generalizes well to new learning tasks. More sophisticated meta-learners augment this procedure by additionally modulating the inner-loop learning dynamics [35, 32, 62, 12, 61, 9].

It has been recently shown that applying MAML while adapting only last-layer weights leads to almost no decrease in performance in standard few-shot learning benchmarks. Our study builds upon the surprising effectiveness of this form of meta-learning, known as almost no inner-loop training [ANIL; 44]. Here, instead of deciding which weights to freeze a priori, we endow the meta-learner with the possibility to explicitly stop changing certain weights in the inner-loop learning process. We do this by introducing an adjustable binary mask which is elementwise multiplied with gradient updates. This can be understood as a simple form of learned gradient modulation that induces sparsity. Overfitting can thus be prevented and learning sped up by focusing adaptation to a sparse parameter subset, discovered by meta-learning.

We find that our sparse-MAML algorithm recovers a behavior that is reminiscent of ANIL. It induces high gradient sparsity in earlier layers of the network while allowing for adaptation in deeper layers including the network's output. Despite this reduction in the number of adaptable parameters, the sparse learning patterns formed by sparse-MAML do not overly specialize to the family of tasks

35th Conference on Neural Information Processing Systems (NeurIPS 2021).

observed during meta-learning; both ANIL and MAML are outperformed by the resulting sparse learners in cross-adaptation problems involving a shift in task distribution [8, 40]. Furthermore, sparsity adapts intuitively to the number of inner-loop gradient steps as well as its learning rate, few-shot dataset size, and network specifications. This leads to a robust and interpretable variant of MAML that improves generalization by self-regularizing the parameters that the model should learn.

An exciting avenue of meta-learning research concerns continual learning. Learning tasks sequentially by gradient descent generally leads to poor results, as past tasks tend to be rapidly forgotten due to interfering weight updates. Such interference can be reduced with online meta-learning methods which optimize the base learning algorithm using both present and past data, kept in a replay buffer [49, 15]. Our findings translate to this setting. We analyze the state-of-the-art look-ahead MAML algorithm [La-MAML; 15] which introduces per-parameter meta-learned learning rates and find that sparse learning emerges, as a large fraction of learning rates drops to zero. Notably, similarly high performance can be reached when meta-learning binary gradient masks only. Moreover, performance improves after endowing a version of MAML adapted for online learning [7] with binary gradient masks. Thus, sparse learning can improve generalization, accelerate future learning, and reduce forgetting, and these benefits can be realized within online meta-learning.

## 2 From MAML to sparse-MAML

**MAML.** The MAML algorithm seeks neural network weights $\theta$ from which only a few gradient descent steps suffice to reach high performance on a given task $\tau$, that is assumed to be drawn from a certain distribution $p(\tau)$. Formally, a task is defined by an outer loss function $L_\tau^{\text{out}}$ and an inner loss function $L_\tau^{\text{in}}$. We will later make explicit the form the two loss functions can take depending on the problem being solved. The result of the inner loss minimization is evaluated by the outer loss leading to the following optimization problem:

$$\min_\theta \mathbb{E}_{\tau \sim p(\tau)}\big[L_\tau^{\text{out}}(\phi_{\tau,K}(\theta))\big] \text{ s.t. } \phi_{\tau,k+1} = \phi_{\tau,k} - \alpha \, \nabla_\phi L_\tau^{\text{in}}(\phi_{\tau,k}) \text{ and } \phi_{\tau,0} = \theta, \qquad (1)$$

with $\phi_{\tau,k}$ denoting the task-specific weights after $k$ steps of gradient descent, $\alpha$ the inner-loop learning rate and $K$ the inner-loop length. The initialization $\theta$ is then obtained by iterative updating, using

$$\theta \leftarrow \theta - \gamma_\theta \, \mathbb{E}_{\tau \sim p(\tau)}\big[\mathrm{d}_\theta \, L_\tau^{\text{out}}(\phi_{\tau,K}(\theta))\big], \qquad (2)$$

with $\gamma_\theta$ the outer-loop learning rate. Note that we need the total derivative $\mathrm{d}_\theta$ in Eq. 2 and not the partial derivative $\nabla_\theta$ due to the complex relationship between $\phi_{\tau,k}$ and $\theta$. In practice, the expectations over the task distribution that appear above are estimated by Monte Carlo integration. The updates in $\theta$ therefore correspond to stochastic gradient descent on the expected outer loss.

In MAML, the total derivative w.r.t. to $\theta$ is obtained by backpropagating through the inner optimization, a resource-intensive procedure. First-order MAML (FOMAML) drastically reduces the computational cost by setting to zero the second-order derivatives that appear when differentiating the inner-loop update.

**Learning the learning rates.** Some variants of MAML focus on learning the learning rate and consider inner-loop updates of the following form:

$$\phi_{\tau,k+1} = \phi_{\tau,k} - \alpha \, M \, \nabla_\phi L_\tau^{\text{in}}(\phi_{\tau,k}), \qquad (3)$$

for some learnable preconditioning matrix $M$, that is optimized similarly to the initialization $\theta$. Through $M$, these algorithms learn some information on the geometry of the loss with the hope of faster inner-loop optimization. Meta-SGD [35] considers a diagonal $M$, i.e., learnable learning rates, meta-curvature [41] considers a block matrix, while vanilla MAML corresponds to the $M = \text{Id}$ case.

**Sparse-MAML.** In line with these approaches, we introduce sparse-MAML. Together with an initial set of weights $\theta$, our algorithm dynamically learns the parameters which will be updated and the ones that will not. Hence, sparse-MAML learns where to learn. To do so, we use a vector $m$ (instead of a matrix $M$) that modulates the gradient in the inner-loop update in the following way:

$$\phi_{\tau,k+1} = \phi_{\tau,k} - \alpha\big(\mathbb{1}_{m \geq 0} \circ \nabla_\phi L_\tau^{\text{in}}(\phi_{\tau,k})\big), \qquad (4)$$

with $\mathbb{1}_{\cdot \geq 0} : \mathbb{R}^n \to \{0,1\}^n$ the step function that is applied elementwise to the underlying parameter vector $m \in \mathbb{R}^n$ and $\circ$ the pointwise multiplication. We differentiate the step function by considering

it linear: this method is called the straight-through estimator [3] and it was recently used for similar large-scale masking [46]. Following FOMAML, we ignore second-order derivatives. This leads to the update

$$m \leftarrow m + \alpha \, \gamma_m \, \mathbb{E}_{\tau \sim p(\tau)} \left[ \nabla_\phi L_\tau^{\text{out}}(\phi_{\tau,K}) \circ \sum_{k=0}^{K-1} \nabla_\phi L_\tau^{\text{in}}(\phi_{\tau,k}) \right]. \tag{5}$$

A detailed derivation of the mask update, alongside the presentation of the initialization update, can be found in the supplementary material (SM).

Our mask update depends on the alignment between the outer-loss gradient $g_\tau^{\text{out}} := \nabla_\phi L_\tau^{\text{out}}(\phi_{\tau,K})$ and the inner loss gradient $\bar{g}_\tau^{\text{in}} := \sum_{k=0}^{K-1} \nabla_\phi L_\tau^{\text{in}}(\phi_k)$ accumulated over the inner loop trajectory. Learning tends to be shut off on coordinates $i$ for which these two quantities are of opposing sign, $\mathbb{E}_\tau \left[ g_{\tau,i}^{\text{out}} \, \bar{g}_{\tau,i}^{\text{in}} \right] < 0$. Such freezing of learning when parameter updates are conflicting on the training and validation sets can decrease negative interference across tasks, which can in turn improve generalization performance [39, 49].

# 3  Few-shot learning

Finding a network that performs well when trained on few samples of unseen data can be formulated as a meta-learning problem. We study here the supervised few-shot learning setting where tasks comprise small labelled datasets. A loss function $\mathcal{L}(\phi, \mathcal{D})$ measures how much the predictions of a network parameterized by $\phi$ deviate from the ground truth labels on dataset $\mathcal{D}$. During meta-learning, the data of a given task $\tau$ is split into training and validation datasets, $\mathcal{D}_\tau^{\text{t}}$ and $\mathcal{D}_\tau^{\text{v}}$, respectively. The sparse-MAML formulation of few-shot learning then consists in optimizing the meta-parameters $\theta$ and $m$ that, given the training set, in turn yield parameters $\phi$ that improve validation set performance:

$$\min_\theta \; \mathbb{E}_{\tau \sim p(\tau)}[\mathcal{L}(\phi_{\tau,K}(\theta, m), \mathcal{D}_\tau^{\text{v}})]$$
$$\text{s.t.} \quad \phi_{\tau,k+1} = \phi_{\tau,k} - \alpha \, \mathbb{1}_{m \geq 0} \circ \nabla_\phi \mathcal{L}(\phi_{\tau,k}, \mathcal{D}_\tau^{\text{t}}) \; \text{ and } \phi_{\tau,0} = \theta, \tag{6}$$

This corresponds to setting the outer- and inner-loop loss functions introduced in Section 2 to $L_\tau^{\text{out}}(\phi) = \mathcal{L}(\phi, \mathcal{D}_\tau^{\text{v}})$ and $L_\tau^{\text{in}}(\phi) = \mathcal{L}(\phi, \mathcal{D}_\tau^{\text{t}})$.

We apply sparse-MAML to the standard few-shot learning benchmark based on the miniImageNet dataset [47]. Our main purpose is to understand whether our meta-learning algorithm gives rise to sparse learning by shutting off weight updates, and if the resulting sparse learners achieve better generalization performance. Furthermore, we analyze the patterns of sparsity discovered by sparse-MAML over a range of hyperparameter settings governing the meta-learning process.

Our experimental setup[1] follows refs. [11, 56] unless stated otherwise. In particular, by default, our experimental results are obtained using the standard 4-convolutional-layer neural network (ConvNet) model that has been intensively used to benchmark meta-learning algorithms. As is also conventional,

---

[1]Source code available at: `https://github.com/Johswald/learning_where_to_learn`

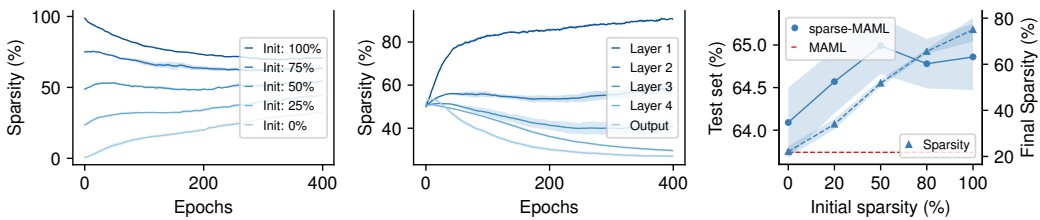

Figure 1: Gradient sparsity emerges in 5-shot, 5-way miniImageNet classification, standard ConvNet model. Results averaged over 5 seeds $\pm$ std. *Left*: Averaged gradient sparsity adapts for different sparsity initializations. *Center*: Different levels of gradient sparsity for convolutional and output layer weights emerge, with gradually less sparsity from earlier to deeper layers, while all being initialized at $\sim 50\%$ sparsity. *Right:* Sparse-MAML reaches higher test set accuracy for higher initial levels of gradient sparsity.

we consider two data regimes: 5-shot 5-way, and 1-shot 5-way (the term 'shot' denotes the number of examples per class, and 'way' the number of classes). As we vary the hyperparameters of our algorithms, we monitor few-shot learning performance and the gradient sparsity level, defined for a parameter group or the entire network as $\|\mathbb{1}_{m<0}\|^2/\dim(m)$. All experimental details can be found in the SM.

## 3.1 Gradient sparsity decreases with layer depth

Our first finding validates and extends the phenomena described by Raghu et al. [44] and Chen et al. [9]. As shown in Figure 1, sparse-MAML dynamically adjusts gradient sparsity across the network, with very different values over the layers. As an example, we show the average gradient sparsity of the four convolutional weight matrices and the output layer during training. The same trend is observed for other parameter groups in the network except the output bias (for which sparsity is always high; see SM). Sparsity clearly correlates with depth and gradually increases towards the early layers of the network, despite the similar value before training (around 50%), i.e., sparse-MAML suppresses inner-loop updates of weights in earlier layers while allowing deeper layers to adjust to new tasks. This effect is robust across different sparsity initializations, with final few-shot learning performance correlating with sparsity, cf. Figure 1.

Table 1: 5-way few-shot classification accuracy (%) on miniImageNet, standard ConvNet model. We report mean $\pm$ std. over 5 seeds. All results except ours taken from the respective papers (we use the symbol '—' to indicate missing results). The results for meta-curvature (MC) are not directly comparable as additional data augmentation was used.

| Method | 1-shot | 5-shot |
|---|---|---|
| MAML [11] | $48.07^{\pm1.75}$ | $63.15^{\pm0.91}$ |
| ANIL [44] | $46.70^{\pm0.40}$ | $61.50^{\pm0.50}$ |
| BOIL [40] | $49.61^{\pm0.16}$ | $66.45^{\pm0.37}$ |
| Meta-SGD [35] | $50.47^{\pm1.87}$ | $64.03^{\pm0.94}$ |
| MT-net [32] | $51.70^{\pm1.84}$ | — |
| MC (+data aug.) [41] | $54.23^{\pm0.88}$ | $68.47^{\pm0.69}$ |
| Shrinkage [9] | $47.7^{\pm0.5}$ | — |
| exp-MAML | $48.38^{\pm0.45}$ | $65.21^{\pm0.62}$ |
| sparse-ReLU-MAML | $49.84^{\pm0.49}$ | $66.80^{\pm0.43}$ |
| sparse-MAML | $50.35^{\pm0.39}$ | $67.03^{\pm0.74}$ |
| sparse-MAML$^+$ | $51.04^{\pm0.59}$ | $68.05^{\pm0.84}$ |

These findings validate that our method can discover sparse learning algorithms. Moreover, they show that the level of sparsity is anti-correlated with depth. This result can be interpreted in the light of neural network models with human-engineered patterns of frozen features, which freeze layers of features based on their depth (in combination with MAML, see e.g. ANIL and BOIL, [44, 40]). Our method justifies these approaches, while outperforming them, cf. Table 1, suggesting that it might be preferable to meta-learn which features to freeze. We note that another related method for automatic discovery of task-shared weights based on learning per-parameter $L_2$ regularization strengths [Shrinkage, 9] yields a similar trend of high freezing for lower-level features, without however improving performance against standard MAML. Our findings hold when applying our method to a deeper and wider residual neural network (ResNet-12) model, see Tables 2 and S1 (SM), where we observe the same trend of decreasing gradient sparsity with depth emerge.

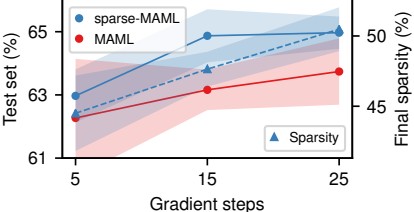 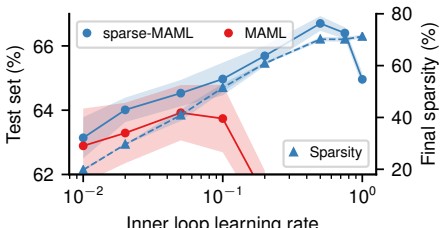

Figure 2: Sparse learning algorithms found by sparse-MAML work best in combination with highly-plastic models. Both gradient sparsity and generalization performance increase with number of inner-loop learning steps (*left*) and learning rate (*right*). Standard MAML, which does not employ sparse learning, requires more careful learning rate tuning and does not benefit as much from large learning rates. In all experiments, gradient sparsity is initially $\sim 50\%$. The inner-loop learning rate is set to 0.1 when varying the number of steps. Results are for 5-shot, 5-way miniImageNet, averaged over 5 seeds $\pm$ std.

## 3.2 Sparse learning prefers highly-plastic models

We hypothesize that restricting learning to an appropriate parameter subset allows for longer training and larger changes without overfitting, beyond meta-learning initial parameter values. To verify this hypothesis we scan over different inner-loop learning rates and lengths and compare the resulting test set performances of MAML and sparse-MAML.

First, we test three different inner-loop durations (5, 15 or 25 gradient steps, see Figure 2, left). We find that neither MAML nor sparse-MAML exhibit overfitting for the duration range considered here (for reference, the original study of MAML applied 5 inner-loop steps during meta-training). In contrast to MAML, the solutions found by sparse-MAML generalize significantly better for longer adaptation phases. This improvement in generalization performance is accompanied by an increase in gradient sparsity. Furthermore, applying sparse-MAML in the very-low data regime of 1-shot learning results in higher levels of gradient sparsity, even though the exact same model and training setup is used for both 1- and 5-shot learning experiments.

We further investigate if increasing the learning rate can result in improved generalization performance in combination with sparse learning. We scan the inner-loop learning rate over a large range, cf. Figure 2 (right), and find a clear trend towards gradient sparsity going along with better test-set accuracy for larger learning rates. Interestingly, similar effects have been reported in standard (non-meta-learned) neural network training where both freezing layers throughout training [45, 5] and the use of large learning rates [33] seem to improve generalization performance.

## 3.3 Sparse learning vs. more expressive gradient modulation methods

Sparse-MAML can be understood as a binary gradient modulation method. Second-order methods such as meta-curvature [41] modulate gradients by meta-learning preconditioning matrices; in meta-SGD [35], these matrices are restricted to be diagonal; sparse-MAML further restricts the diagonal values to be binary. From this point of view, sparse-MAML is the least expressive form of gradient modulation.

Surprisingly, we find that despite its reduced expressiveness, sparse-MAML recovers the performance improvements achieved by the more sophisticated alternatives, significantly improving the performance of standard MAML (cf. Table 1). We point out that sparse-MAML uses a

Table 2: 5-way few-shot classification accuracy (%) on miniImageNet with a ResNet-12 model. We report mean $\pm$ std. over 3 seeds. We report MetaOptNet [31] figures when no additional regularization techniques are applied. Results from BOIL and MetaOptNet are taken from the respective papers.

| Method | 1-shot | 5-shot |
|---|---|---|
| MetaOptNet | 51.13 | 70.88 |
| MAML | $53.91^{\pm0.61}$ | $69.36^{\pm1.23}$ |
| ANIL | $55.25^{\pm0.33}$ | $70.03^{\pm0.58}$ |
| BOIL | — | $70.50^{\pm0.28}$ |
| sparse-MAML | $55.02^{\pm0.46}$ | $70.02^{\pm1.12}$ |
| sparse-ReLU-MAML | $56.39^{\pm0.38}$ | $73.01^{\pm0.24}$ |

first-order update (Eq. 5), while all three gradient modulation methods we compare to (meta-SGD, meta-curvature and MT-nets) use second-order derivatives that are more costly to evaluate.

**Sparse learning emerges when meta-learning learning rates.** We also implement a variant of meta-SGD which uses rectified learning rates (sparse-ReLU-MAML). Concretely, we replace the step function $\mathbb{1}_{m\geq0}$ in the inner-loop dynamics (Eq. 4) by the positive part of $m$, $(m)_+ := \mathbb{1}_{m\geq0} m$. Then, we learn the underlying learning rate parameter $m$ using our first-order straight-through update of Eq. 5 to

Table 3: Average gradient sparsity levels (%) after meta-learning on 5-way miniImageNet few-shot tasks, standard ConvNet model. Mean $\pm$ std. over 5 seeds.

| Method | 1-shot | 5-shot |
|---|---|---|
| sparse-ReLU-MAML | $77.53^{\pm0.73}$ | $73.53^{\pm0.85}$ |
| sparse-MAML | $79.04^{\pm1.61}$ | $74.98^{\pm0.10}$ |
| sparse-MAML$^+$ | $78.05^{\pm1.67}$ | $76.66^{\pm1.13}$ |

prevent learning rates from getting stuck at zero. Besides standard meta-SGD, which allows learning rates to go negative, we compare this method to an alternative exponential learning rate parameterization [53], $\exp m$, which like sparse-ReLU-MAML enforces non-negativity while avoiding permanently frozen updates (exp-MAML, Table 1). It is, however, harder to reach sparse learning rate

distributions under this parameterization, as the meta-gradient $d_m \mathcal{L}$ becomes exponentially small as $m$ approaches zero.

We analyze the distributions of learning rates that sparse-ReLU-MAML yields on miniImageNet and observe that gradient sparsity once more emerges, cf. Table 3. We find that the levels of gradient sparsity and generalization performance when meta-learning binary (sparse-MAML) or rectified learning rates (sparse-ReLU-MAML) are approximately the same, with both methods outperforming exp-MAML on both 1-shot and 5-shot tasks. These results support the hypothesis that shutting off weight updates is one of the essential gradient modulation operations in few-shot learning. We note that while sparse-MAML and sparse-ReLU-MAML quickly disable learning in a large fraction of weights, exp-MAML tends to push learning rates down, in particular for layers close to the input, but at a much slower pace; increasing the meta-learning rate $\gamma_m$ cannot compensate for this slowdown as learning becomes unstable (data not shown).

This picture changes for the the deeper and larger ResNet-12 model, cf. Table 2. When using this more complex architecture, we find that sparse rectified learning rates (sparse-ReLU-MAML) are beneficial over binary gradient masks (sparse-MAML). In particular, the combination of sparse learning (see Table S1 for additional details) with learning rate modulation found by sparse-ReLU-MAML outperforms all other methods, including standard (dense-learning) MAML, as well as methods based on manually freezing layers in the inner-loop: BOIL [40], ANIL [44], and the closely-related MetaOptNet [31] method. Like ANIL, MetaOptNet only adapts the final classification layer in the inner-loop, but it uses a more sophisticated solver instead of a few steps of gradient descent to learn task-specific solvers. Thus, once more, learning by sparse gradient descent is an effective strategy to improve the generalization performance of a few-shot learner.

**Stochastic gradient masking.** We further investigate whether stochastic binary gradient masks can improve few-shot learning performance. Our interest in studying stochastic masks is two-fold: as a way to improve meta-optimization based on our straight-through estimator; and to determine if stochastic masking is beneficial at meta-test time. We thus investigate sparse-MAML$^+$, a variant of our algorithm in which gradient masks are generated from a low-dimensional Gaussian vector, with noise intensity determined by meta-learning (see SM). As before, we adjust meta-parameters using a first-order update. We find that this mask generation method does result in improved performance, cf. Table 1. Interestingly, mask randomness is entirely suppressed by meta-learning; eventually, $\sigma \to 0$, and we recover a single deterministic mask $m$. The performance improvements observed on few-shot learning therefore stem from improvements to the meta-optimization process, likely related to the challenges of optimizing binary variables with (pseudo)gradient-based methods.

### 3.4 Sparse learning improves performance in cross-domain adaptation tasks

We now investigate whether the patterns of gradient sparsity discovered by our method overfit to the particular task family where they were obtained, namely, to few-shot miniImageNet classification

Table 4: Few-shot classification accuracy (%) when meta-learning on miniImageNet but meta-testing on TieredImageNet, CUB and Cars. Mean $\pm$ std. over 5 seeds.

| Problem | Method | TieredImageNet | CUB | Cars |
|---|---|---|---|---|
| 1-shot | MAML | $51.61^{\pm0.20}$ | $40.51^{\pm0.08}$ | $33.57^{\pm0.14}$ |
| | ANIL | $52.82^{\pm0.29}$ | $41.12^{\pm0.15}$ | $34.77^{\pm0.31}$ |
| | BOIL | $53.23^{\pm0.41}$ | $\mathbf{44.20}^{\pm0.15}$ | $36.12^{\pm0.29}$ |
| | sparse-ReLU-MAML | $53.18^{\pm0.52}$ | $41.86^{\pm0.95}$ | $35.46^{\pm0.67}$ |
| | sparse-MAML | $53.47^{\pm0.53}$ | $41.37^{\pm0.73}$ | $35.90^{\pm0.50}$ |
| | sparse-MAML$^+$ | $\mathbf{53.91}^{\pm0.67}$ | $43.43^{\pm1.04}$ | $\mathbf{37.14}^{\pm0.77}$ |
| 5-shot | MAML | $65.76^{\pm0.27}$ | $53.09^{\pm0.16}$ | $44.56^{\pm0.21}$ |
| | ANIL | $66.52^{\pm0.28}$ | $55.82^{\pm0.21}$ | $46.55^{\pm0.29}$ |
| | BOIL | $69.37^{\pm0.23}$ | $60.92^{\pm0.11}$ | $50.64^{\pm0.22}$ |
| | sparse-ReLU-MAML | $69.06^{\pm0.28}$ | $59.55^{\pm1.23}$ | $51.21^{\pm0.89}$ |
| | sparse-MAML | $68.83^{\pm0.65}$ | $60.58^{\pm1.10}$ | $52.63^{\pm0.56}$ |
| | sparse-MAML$^+$ | $\mathbf{69.92}^{\pm0.21}$ | $\mathbf{62.02}^{\pm0.78}$ | $\mathbf{53.18}^{\pm0.44}$ |

tasks. This is an important question, since excessive parameter freezing may prevent adaptation to tasks that are too different from those presented during meta-learning.

We therefore move our analysis of few-shot learning to a cross-domain adaptation setting. In cross-domain adaptation problems, the family of tasks presented post-meta-learning to evaluate our algorithms is shifted by sampling classes from a different dataset. In particular, we train our meta-learner on the miniImageNet dataset and then evaluate learning performance on the TieredImageNet, CUB and Cars datasets. It has previously been demonstrated that manually freezing either the head (BOIL) or the body (ANIL) during meta-testing improves performance in this setting [40], compared to letting all weights adapt (MAML). In Table 4 we compare the performance of our method to these baselines. We find that meta-learning the freezing pattern with sparse-MAML as opposed to manually selecting it consistently improves cross-domain adaptation.

# 4 Continual learning

We now turn to a continual learning setting, where tasks must be learned sequentially. A successful continual learner is able to learn similar tasks faster, as in the few-shot learning case, while retaining high performance on previously seen tasks. We conjecture that sparse learning can improve memory retention and accelerate future learning by reducing interference with past updates.

## 4.1 Gradient sparsity emerges when learning continually with Look-ahead MAML

We investigate the benefits of sparse gradients in the recently proposed La-MAML algorithm [15]. This algorithm combines online meta-learning in conjunction with a small replay buffer which holds representative examples from the past in memory. Standard replay methods [51], define a joint objective using present and buffered data and directly optimize this objective. La-MAML follows a technique known as meta-experience replay [49] and introduces a bi-level optimization problem. The outer loss $L^{\text{out}}$ is the multi-task objective optimized with standard replay methods, while the inner loss $L^{\text{in}}$ is evaluated on the new incoming data only. Riemer et al. [49] have shown that such meta-learning promotes gradient alignment over tasks, which is a way to reduce interference [36].

Like the variants of MAML reviewed in Section 2, La-MAML introduces meta-learned per-parameter learning rates. We now briefly review a single iteration of the algorithm; complete pseudocode is provided in the SM. Each iteration of La-MAML consists of processing a new batch of data $\mathcal{B}$ as follows: (*i*) starting from $\phi_0 = \theta$ taking an inner-loop step on each sample $k$ in $\mathcal{B}$, with $L_k^{\text{in}}(\phi_k) = \mathcal{L}(\phi_k, \mathcal{B}_k)$; (*ii*) defining an outer-loss $L^{\text{out}}(\phi_K, \mathcal{B} \cup \mathcal{R})$ on both new data $\mathcal{B}$ and a batch of past data $\mathcal{R}$ sampled from the replay buffer; (*iii*) taking an outer-loop step on the learning rate parameter using a first-order update followed by an outer-loop step (using the newly updated learning rate) on the neural network parameters $\theta$; (*iv*) re-populating the replay buffer with data in $\mathcal{B}$. The

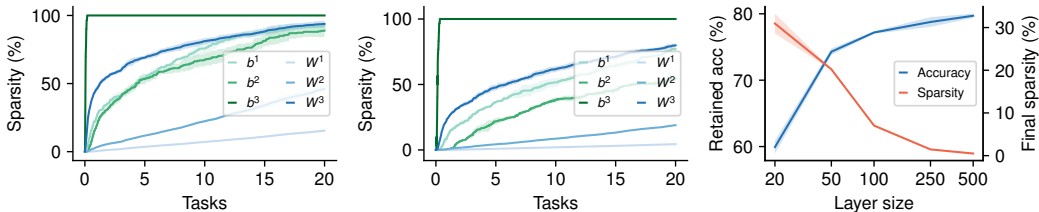

Figure 3: Gradient sparsity when learning MNIST rotations with the La-MAML and sparse-La-MAML algorithms. Results averaged over 3 seeds $\pm$ std. *Left*: Sparsity emerges on the original La-MAML algorithm across the three layer network and monotonically increases with the number of tasks and with depth for both weight ($W_1, W_2, W_3$) and bias parameters ($b_1, b_2, b_3$). *Center*: A similar behavior is observed when replacing meta-learned learning rates by meta-learned binary gradient masks (sparse-La-MAML). *Right:* Overall sparsity of sparse-La-MAML decreases with increased network capacity accompanied with higher retained accuracy (RA). Network capacity is varied by changing the number of neurons in the two hidden layers simultaneously.

sequence of inner-loop updates is given by

$$\phi_{k+1} = \phi_k - (\alpha)_+ \circ \nabla_\phi L_k^{\text{in}}(\phi_k), \qquad \text{s.t. } \phi_0 = \theta, \tag{7}$$

where $\alpha$ is a vector of learning rates whose components are constrained to be non-negative by elementwise application of the positive part function. We note that while the main text of ref. [15] presents an inner-loop learning rate parameter that is allowed to go negative, the implementation for the experiments reported in ref. [16] uses rectified learning rates. In this implementation, a learning rate that is updated below zero will never recover, which can lead to dead coordinates and promote sparsity. The inner loss $L_k^{\text{in}}(\phi)$ is defined on a different data sample on each step $k$. A first-order update is applied to $\theta$, again modulated by the adaptive learning rate:

$$\theta \leftarrow \theta - (\alpha)_+ \circ \nabla_\phi L^{\text{out}}(\phi_K). \tag{8}$$

**Sparse-La-MAML.** Our sparse-MAML can be readily applied to continual learning problems by modifying the inner- and outer-loop updates of La-MAML. We replace the meta-learned learning rates in equations 7-8 by meta-learned binary gradient masks, $\alpha = \alpha_0 \mathbb{1}_{m \geq 0}$, with $\alpha_0 \in \mathbb{R}_+$ some scalar (fixed) learning rate value. To learn the underlying parameter $m$, we again resort to our first-order update (equation 5).

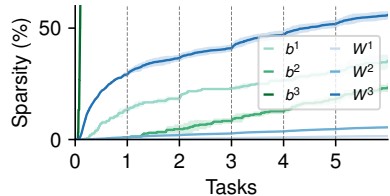

Figure 4: Structured sparsity emerges and tends to converge within a task in multi-pass continual learning. Results shown for La-MAML after training on MNIST rotations, averaged over 3 seeds $\pm$ std., for weight layers $(W_1, W_2, W_3)$ and bias parameters $(b_1, b_2, b_3)$. Sparsity increases with depth.

**Sparse learning improves continual learning.** We hypothesize that a large fraction of learning rates approaches zero when the hyperparameters of La-MAML and sparse-La-MAML are tuned for best continual learning performance. To test this hypothesis, we follow the exact same setup as in the original study of La-MAML [15]. We perform a grid search over the learning rates $\alpha_0$ and $\gamma_m$, and search for best continual learning performance, not sparsity (cf. SM). The remaining hyperparameters are kept to the values provided in [15].

We study the three MNIST [30] continual learning problems *rotations*, *permutations* and *many permutations* using a single-headed network using the code accompanying ref. [15]. Task information is not given to the network, and each data point is seen only once, unless noted otherwise. Full details as well as additional experiments using the CIFAR-10 [28] dataset are provided in the SM.

We verify that our initial hypothesis is correct: La-MAML shuts off learning in many coordinates (cf. Figure 3; full results may be found in the SM, Table S5), reaching even higher levels of sparsity than sparse-La-MAML. This can be explained by the fact that dead coordinates can arise in La-MAML, which can lead to excess sparsity. By contrast, our straight-through update dynamically and continually adjusts the pattern of sparsity allowing previously frozen parameters to be unfrozen. This results in matching or slightly improved performances when using our binary gradient mask across all three MNIST variants, see Table 5, both in terms of final retained accuracy (RA) and backward-transfer and interference (BTI; the change in accuracy measured at the end of the experiment minus just after learning a task, averaged over tasks). Moreover, the patterns of sparsity adjust to the capacity of the network, decreasing and eventually vanishing for larger models (Figure 3), as retained accuracy goes up, indicating that the task is not sufficiently difficult to create interference on large capacity models.

As in our few-shot learning experiments, structured sparsity emerges across the different parameter groups of the network (cf. Figure 3). We observe that now sparsity is highest closest to the output layer, the exact opposite of the trend found in our few-shot learning experiments. This provides evidence that online meta-learning can discover how to rewire low-level features without interference in order to accommodate different tasks that share high-level structure. We further investigate a multi-pass setting, where the examples from each task are visited multiple times (10 epochs instead of 1) before proceeding to the next task. In this setting, it can be seen that sparsity levels (displayed in Figure 4) tend to converge within tasks and then raise again when tasks switch, presumably to preserve past memories via gradient sparsification. Taken together, our results support the hypothesis that gradient sparsity is beneficial for continual learning and that appropriate patterns of sparsity can be discovered by simple online gradient-based meta-learning.

Table 5: Retained accuracy (RA) and backward-transfer and interference (BTI) for three different MNIST continual learning problems: rotations, permutations and many permutations. We report mean ± std. over 5 seeds. Negative BTI values closer to zero imply less forgetting and are therefore better. Results of related work are taken from [15]; for completeness we include the GEM [36] and MER [49] methods next to a stochastic gradient descent baseline. Although sparse-La-MAML (sp-LaM) is strictly less expressive than the original La-MAML algorithm, it shows competitive performance across all variants and both metrics. The lower baseline BTI values can be explained by lower overall accuracies achieved by La-MAML.

| Method | Rotations | | Permutations | | Many permutations | |
|---|---|---|---|---|---|---|
| | RA | BTI | RA | BTI | RA | BTI |
| Baseline | $53.38^{\pm1.53}$ | $\mathbf{-5.44}^{\pm1.70}$ | $55.42^{\pm0.65}$ | $-13.76^{\pm1.19}$ | $32.62^{\pm0.43}$ | $-19.06^{\pm0.86}$ |
| GEM | $67.38^{\pm1.75}$ | $-18.02^{\pm1.99}$ | $55.42^{\pm1.10}$ | $-24.42^{\pm1.10}$ | $32.14^{\pm0.50}$ | $-23.52^{\pm0.87}$ |
| MER | $77.42^{\pm0.78}$ | $-5.60^{\pm0.70}$ | $73.46^{\pm0.45}$ | $-9.96^{\pm0.45}$ | $47.40^{\pm0.35}$ | $-17.78^{\pm0.39}$ |
| La-M | $77.42^{\pm0.65}$ | $-8.64^{\pm0.40}$ | $74.34^{\pm0.67}$ | $\mathbf{-7.60}^{\pm0.51}$ | $48.46^{\pm0.45}$ | $\mathbf{-12.96}^{\pm0.07}$ |
| sp-LaM | $\mathbf{77.77}^{\pm0.58}$ | $-8.16^{\pm0.61}$ | $\mathbf{76.88}^{\pm0.72}$ | $-8.39^{\pm0.63}$ | $\mathbf{50.81}^{\pm0.79}$ | $-13.73^{\pm0.73}$ |

## 4.2 Sparse online learning

We finally consider another online learning setting in which the underlying task is concealed from the learner and can randomly change at each step, potentially going back to previously seen tasks [50, 19, 7]. At each time step $t$, the data $\mathcal{D}_t$ is an i.i.d. sample from a stationary distribution that only depends on the current task. The learner, whose current state is denoted by $\phi_t$, is evaluated whenever new data is presented and modifies its behavior accordingly. The goal is then to minimize the cumulative loss $\sum_{t=1}^{T} \mathcal{L}(\phi_t, D_t)$ measuring the performance before adaptation takes place.

This online learning protocol differs from the one adopted in the previous section, where tasks were visited only once and only the final loss $\sum_{t=1}^{T} \mathcal{L}(\phi_T, \mathcal{D}_t)$ evaluated at $\phi_T$ mattered. The cumulative loss criterion

Table 6: Sparse learning improves continual-MAML performance. Cumulative online accuracy on Omniglot-MNIST-FashionMNIST benchmark. Tasks switch with probability $1 - p$. Results from previous work taken from [7]. Mean ± std. over 5 seeds.

| Method | $p = 0.98$ | $p = 0.9$ |
|---|---|---|
| Online Adam [25] | $73.9^{\pm2.2}$ | $23.8^{\pm1.2}$ |
| Fine-tuning | $72.7^{\pm1.7}$ | $22.1^{\pm1.1}$ |
| MAML [11] | $84.5^{\pm1.7}$ | $75.5^{\pm0.7}$ |
| ANIL [44] | $75.3^{\pm2.0}$ | $69.1^{\pm0.8}$ |
| BGD [60] | $87.8^{\pm1.3}$ | $63.4^{\pm0.9}$ |
| MetaCOG [19] | $88.0^{\pm1.0}$ | $63.6^{\pm0.9}$ |
| MetaBGD [19] | $91.1^{\pm2.6}$ | $74.8^{\pm1.1}$ |
| C-MAML | $92.8^{\pm0.6}$ | $83.3^{\pm0.4}$ |
| sparse-C-MAML | $94.2^{\pm0.4}$ | $86.3^{\pm0.4}$ |
| sparse-ReLU-C-MAML | $93.5^{\pm0.5}$ | $86.1^{\pm0.2}$ |

emphasizes fast learning and adaptability while memory is still needed to avoid re-learning, since tasks can be re-encountered. Recently, it has been shown that a simple modification of MAML [continual-MAML; 7] can outperform a number of algorithms specifically tailored for this setting as well as plain stochastic gradient descent [4]. Briefly, continual-MAML extends MAML by introducing a task-switch detection mechanism based on changes in loss; data is buffered until a switch is detected. When this occurs, the buffered data is used to perform a meta-parameter update; the buffer is reset; and the inner-loop optimization restarts. Here, we merge continual-MAML with sparse-MAML, and modulate inner-loop gradients according to equation 4. We present complete pseudocode for the algorithm in the SM.

We reproduce the experiments of [7] in which a sequence of 10000 examples from the Omniglot [29], MNIST [30] and FashionMNIST [59] datasets is presented for online learning to a single-headed neural network, using the code provided by the authors. We carry out a grid search to tune the inner-loop learning rate $\alpha_0$ and the mask learning rate $\gamma_m$ introduced by sparse-MAML for best performance, not sparsity (see SM). We observe again structured (layer-dependent) gradient sparsity emerge when using this algorithm (sparse-C-MAML), as shown in Figure S5, and an increase in cumulative online learning accuracy over the original continual-MAML algorithm (cf. Table 6). Finally, we observe the same qualitative behavior (see SM for sparsity levels) and obtain similar performance when replacing our binary masks by rectified learning rates (sparse-ReLU-C-MAML)

meta-learned with our straight-through update. These findings once more support the hypothesis that sparse learning, and not learning rate modulation, lead to the improved performance reported here.

# 5 Discussion

We studied gradient-based meta-learning systems with the ability of learning where to learn. This was modeled by adding binary variables which masked gradients on a per-parameter basis, therefore determining which parameters are allowed to change. We observed gradient sparsity emerge in standard few-shot and continual learning problems, without introducing an explicit bias towards sparsity. This form of sparse learning, which may be understood as sparse gradient descent, was accompanied by overall improvements in generalization, as well as reduced interference and forgetting.

Previous work on gradient modulation has focused on estimating task-shared loss geometry to precondition the optimization procedure [35, 62, 12, 32, 61]. In addition, a stochastic variant of gradient masking was featured in the MT-net algorithm [32] as part of a more complex model. Our approach differs from these previous studies in its simplicity. We restrict gradient modulation to be binary and deterministic and use an inexpensive first-order update to learn the gradient masks. In contrast to traditional methods for inducing sparsity via regularization [54] (here, gradient regularization) our approach does not require evaluating second derivatives, which would result from differentiating gradient regularizers. Despite these simplifications, we find competitive performance on our experiments. These results point towards sparse gradient descent as a powerful learning principle.

The idea of meta-learning learning rates can be traced back to the seminal work of Sutton [53], who proposed to estimate learning rate meta-gradients online using forward-mode automatic differentiation, and to use consecutive batches of data to define inner- and outer-loop loss functions. This approach, known as stochastic meta-descent (SMD), was extended to nonlinear models by Schraudolph [52] using fast Hessian-vector product techniques. Using SMD to optimize neural network models is an ongoing area of research [55, 58, 22, 24]. It is an interesting question whether gradient sparsity emerges when applying SMD to online learning problems that are not clearly structured in tasks, as considered here. Furthermore, this line of work suggests that it might be possible to obtain finer binary gradient mask updates in an online fashion using forward-mode automatic differentiation.

A recent study has put into question whether any useful adaptation still takes place when MAML few-shot learners are presented with a novel task after meta-learning [44]. Our findings shed light on this question, by demonstrating that few-shot learning performance can be improved when learning an adequate small subset of parameters. The additional plasticity of our meta-learned sparse learners led to a significant performance increase over handwired schemes based on frozen layers, in particular when encountering tasks drawn from a different family of problems than that used for meta-learning. This finding complements recent work showing that modular recurrent networks with sparse updating mechanisms hold great promise in improving out-of-distribution generalization performance [14, 38].

Our results may be of special interest to the design of neuromorphic hardware. Updating weights on-chip implies a significant power overhead whose cost scales with the number of plastic weights [42]. Reducing the number of plastic weights can therefore result in immediate improvements in energy efficiency and scalability. Likewise, synaptic plasticity is costly in biological neural networks. Given the high energy demands of the brain there has likely been selective pressure to reduce costs associated with synaptic change [34]. It is therefore conceivable that the brain developed mechanisms to restrict learning to an appropriate subset of synapses to save energy. Our study presents further evidence in favor of sparse synaptic change, given its potential benefits in the biologically-relevant scenarios of few-shot and continual learning investigated here.

**Limitations.** To arrive at our simple mask update (equation 5) we introduced two important approximations. First, we dropped all terms involving second-order derivatives, and second, we used straight-through estimation to differentiate through the step function. Despite their frequent use in previous work, both approximations remain poorly understood. For this reason it is possible that our algorithm fails unexpectedly outside the experiments considered here; potential problems stemming from the non-differentiability of the step function are likely unavoidable in our approach. Our update is also likely inappropriate for long inner loops (large $K$) [58]. Finally, scaling our few-shot learning experiments to more complex neural network models is potentially difficult, a challenge that our approach shares with other methods based on MAML.

## Acknowledgments and Disclosure of Funding

This work was supported by an Ambizione grant (PZ00P3_186027) awarded to João Sacramento by the Swiss National Science Foundation. Johannes von Oswald is funded by the Swiss Data Science Center (J.v.O. P18-03). Dominic Zhao is supported by AlayaLabs (Montreal, Canada). Massimo Caccia was supported through MITACS during his part time employment with Element AI the ServiceNow company, and by Amazon, during his part time employment there. We thank Charlotte Frenkel, Frederik Benzing, Angelika Steger and Laura Sainz for helpful discussions.

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
