# Supplementary Material

# Learning where to learn: Gradient sparsity in meta and continual learning

**Johannes von Oswald\*, Dominic Zhao\*, Seijin Kobayashi, Simon Schug, Massimo Caccia, Nicolas Zucchet, João Sacramento**

## A Derivation of the sparse-MAML update

Here, we derive the sparse-MAML update rules on the initialization $\theta$ and on the underlying mask parameter $m$, that are given by

$$\theta \leftarrow \theta - \gamma_\theta \, \mathbb{E}_{\tau \sim p(\tau)} \left[ \nabla_\phi \, L_\tau^{\text{out}}(\phi_{\tau,K}) \right] \tag{9}$$

$$m \leftarrow m + \alpha \, \gamma_m \, \mathbb{E}_{\tau \sim p(\tau)} \left[ \nabla_\phi \, L_\tau^{\text{out}}(\phi_{\tau,K}) \circ \sum_{k=0}^{K-1} \nabla_\phi \, L_\tau^{\text{in}}(\phi_{\tau,k}) \right]. \tag{10}$$

**Update of the initialization** We first start by deriving the $\theta$-update. To update $\theta$ with gradient descent we need the total derivative $\mathrm{d}_\theta \, L_\tau^{\text{out}}(\phi_{\tau,K})$. Using the chain rule, it is equal to

$$\mathrm{d}_\theta \, L_\tau^{\text{out}}(\phi_{\tau,K}) = \nabla_\phi \, L_\tau^{\text{out}}(\phi_{\tau,K}) \, \mathrm{d}_\theta \, \phi_{\tau,K}.$$

The last term of the right hand side of the previous equation requires backpropagating through the training procedure as modifying the initialization changes the entire trajectory of $\phi$. By using the recursive formulation of $\phi_{\tau,K}$, we have

$$\begin{aligned} \mathrm{d}_\theta \, \phi_{\tau,K} &= \mathrm{d}_\theta \left[ \phi_{\tau,K-1} - \alpha \mathbb{1}_{m \geq 0} \circ \nabla_\phi \, L_\tau^{\text{in}}(\phi_{\tau,K-1}) \right] \\ &= \mathrm{d}_\theta \, \phi_{\tau,K-1} - \alpha \, \mathbb{1}_{m \geq 0} \circ \left( \nabla_\phi^2 \, L_\tau^{\text{in}}(\phi_{\tau,K-1}) \, \mathrm{d}_\theta \, \phi_{\tau,K-1} \right). \end{aligned}$$

In sparse-MAML, we use a first-order approximation that consists in zeroing out all the second order derivatives to keep the computations as simple as possible, while keeping the benefits of meta-learning. It follows that

$$\begin{aligned} \mathrm{d}_\theta \, \phi_{\tau,K} &\approx \mathrm{d}_\theta \, \phi_{\tau,0} \\ &= \mathrm{d}_\theta \, \theta \\ &= \text{Id} \end{aligned}$$

and

$$\mathrm{d}_\theta \, L_\tau^{\text{out}}(\phi_{\tau,K}) \approx \nabla_\phi \, L_\tau^{\text{out}}(\phi_{\tau,K}),$$

leading to the update presented in Eq. 9 once the derivative approximation is inserted in a gradient descent update.

In our online continual learning setting, we additionally apply the mask to the $\theta$-update.

**Update of the mask** The derivation of the underlying mask parameter $m$ update can be done similarly to the one of the $\theta$-update. We first apply the chain rule and get

$$\mathrm{d}_m \, L_\tau^{\text{out}}(\phi_{\tau,K}) = \nabla_\phi \, L_\tau^{\text{out}}(\phi_{\tau,K}) \, \mathrm{d}_m \phi_{\tau,K}.$$

We then compute the derivative of $\phi_{\tau,K}$ with respect to $m$:

$$\mathrm{d}_m \, \phi_{\tau,K} = \mathrm{d}_m \, \phi_{\tau,K-1} - \alpha \, \mathrm{d}_m \left[ \mathbb{1}_{m \geq 0} \circ \nabla_\phi \, L_\tau^{\text{in}}(\phi_{\tau,K-1}) \right].$$

As for the $\theta$-update, we do not take in account second-order derivatives, we thus consider first-order derivatives to be constant. The following terms remain

$$\mathrm{d}_m \, \phi_{\tau,K} \approx \mathrm{d}_m \, \phi_{\tau,K-1} - \alpha \, \mathrm{d}_m \left[ \mathbb{1}_{m \geq 0} \right] \, \text{diag} \left( \nabla_\phi \, L_\tau^{\text{in}}(\phi_{\tau,K-1}) \right).$$

We approximate $\mathrm{d}_m \mathbb{1}_{m \geq 0}$ using straight-through estimation, which consists in taking this derivative equal to the identity, thus having

$$\mathrm{d}_m \, \phi_{\tau,K} \approx \mathrm{d}_m \phi_{K-1} - \alpha \, \mathrm{diag}\big(\nabla_\phi L_\tau^{\mathrm{in}}(\phi_{\tau,K-1})\big)$$

and

$$\mathrm{d}_m \, \phi_{\tau,K} \approx -\alpha \sum_{k=0}^{K-1} \mathrm{diag}\big(\nabla_\phi L_\tau^{\mathrm{in}}(\phi_{\tau,k})\big).$$

Combining everything into a gradient descent update yields the update of Eq. 10.

Note that the updates for $\theta$ and $m$ differ in their structure although both are obtained using first-order approximations. This is because $\theta$ only enters the first update step of $\phi$, while $m$ consistently appears along the whole trajectory of $\phi$.

# B  Additional experimental details and analyses

## B.1  Few-shot learning experiments

### B.1.1  Reproducibility

Unless specified otherwise, all experiments presented in our paper follow the supervised few-shot learning setup studied in ref. [11] and are performed on the miniImageNet dataset [47, 56] which consists of 64 training classes, 12 validation classes and 24 test classes. The backbone classifier consists of four convolutional layers each with 64 filters followed by a batch normalization layer [21] as well as a max-pooling layer with kernel size and stride of 2. The network then projects to its output via a fully-connected layer. We choose to use the 64-filter version (instead of the 32) to be one-to-one comparable to BOIL [40] (and the ANIL results within) which uses the 64 channel variant.

In order to produce the results visualized in Figures 1 and 2, we used the following hyperparameters:

- Batch size 4 and 2 for 1-shot resp. 5-shot experiments (note that BOIL uses 4 for both).
- Inner-loop length $K = 25$ during meta-training and meta-test train.
- Inner-loop learning rate $\alpha = 0.1$.
- Optimizer: Adam with default PyTorch hyperparameters and a learning rate of 0.001 (for meta-parameters $\theta$ and $m$).
- Initialization: Kaiming [18] for meta-parameters $\theta$ and $m$.

Note that when analyzing the effects of varying a particular set of hyperparameters (e.g., the inner-loop learning rate), we hold all other hyperparameters fixed.

We train all models for 400 epochs (600 for sparse MAML$^+$) of 100 training tasks each. In the case of sparse-ReLU we initialize all learnable inner-loop learning rates at the same value $\alpha$, cf. Table S1. Note that this leads to an initial gradient sparsity level of $0\%$, while still converging to high sparsity levels.

All our few-shot learning results are reported for models that are early-stopped by measuring the average validation set accuracy (across 300 validation set tasks). The model with best average validation set accuracy is then tested on 300 tasks of the test set data and the cross-domain datasets.

We handle batch normalization parameters following the *transductive* learning setting, as originally done in MAML [11, 39].

For the results shown in Table 1, we tuned the best values found by scanning over learning rates and inner-loop lengths using a sparsity initialization of $50\%$. Additional details can be found in Table S2.

**ResNet-12**  For the ResNet-12 results shown in Table 2, we tuned the best values found by scanning over inner-loop learning rates and inner-loop lengths with a sparsity initialization of $50\%$. For sparse-ReLU-MAML we initialized the inner-loop learning rate to be $\alpha$ without any randomness. For all experiments we optimize meta-parameters with Adam [25]. We also set $\gamma_\theta = 0.001, \alpha = 0.05$ and $\gamma_m = 0.01, K_{\mathrm{test/train}} = 35$. Meta-gradients are clipped to lie within $[-10, 10]$. The architecture is identical to the one used in previous meta-learning studies [40, 31]. We tested two different sizes

Table S1: Detailed results for the large and small ResNet-12 models on miniImageNet 5-way 1- and 5-shot experiments, including sparsity levels and the performance of an economical snapshot ensemble method used in previous studies of MAML [2]. Mean ± std. over 3 seeds.

| Arch. | Problem | Method | Acc. (%) | Ensemble Acc. (%) | Sparsity (%) |
|---|---|---|---|---|---|
| Large | 1-shot | MAML | $53.51^{\pm 1.24}$ | $55.65^{\pm 0.81}$ | — |
| | | ANIL | $52.95^{\pm 1.30}$ | $55.23^{\pm 0.66}$ | all except head |
| | | sp-M | $55.18^{\pm 0.50}$ | $56.83^{\pm 0.08}$ | 48.39 |
| | | sp-ReLU-M | $55.29^{\pm 0.56}$ | $57.44^{\pm 0.43}$ | 29.56 |
| | 5-shot | MAML | $69.58^{\pm 1.08}$ | $72.77^{\pm 0.60}$ | — |
| | | ANIL | $69.39^{\pm 1.28}$ | $73.07^{\pm 0.42}$ | all except head |
| | | sp-M | $69.93^{\pm 0.61}$ | $72.83^{\pm 0.35}$ | 23.57 |
| | | sp-ReLU-M | $72.93^{\pm 0.92}$ | $75.60^{\pm 0.12}$ | 12.95 |
| Small | 1-shot | MAML | $53.91^{\pm 0.61}$ | $56.09^{\pm 0.12}$ | — |
| | | ANIL | $55.25^{\pm 0.33}$ | $57.02^{\pm 0.21}$ | all except head |
| | | sp-M | $55.02^{\pm 0.46}$ | $57.53^{\pm 0.25}$ | 37.56 |
| | | sp-ReLU-M | $56.39^{\pm 0.38}$ | $58.41^{\pm 0.38}$ | 28.44 |
| | 5-shot | MAML | $69.36^{\pm 0.23}$ | $72.50^{\pm 0.22}$ | — |
| | | ANIL | $70.03^{\pm 0.58}$ | $73.09^{\pm 0.13}$ | all except head |
| | | sp-M | $70.02^{\pm 1.12}$ | $72.87^{\pm 0.59}$ | 15.09 |
| | | sp-ReLU-M | $73.01^{\pm 0.24}$ | $75.52^{\pm 0.48}$ | 15.78 |

Table S2: Hyperparameters of sparse-MAML, sparse-MAML$^+$ and sparse-ReLU-MAML to obtain the reported results for in- and cross dataset few-shot experiments.

| Problem | Method | Optimizer | $K_{\text{train}}$ | $K_{\text{test}}$ | $\alpha$ | $\gamma_m$ | $K_{\text{test}}^{\text{tiered}}/K_{\text{test}}^{\text{CUB}}/K_{\text{test}}^{\text{Cars}}$ |
|---|---|---|---|---|---|---|---|
| 1-shot | sp-M | Adam | 35 | 100 | 0.25 | 0.0075 | 35 |
| | sp-M$^+$ | SGD+N | 35 | 100 | 0.1 | 0.0075 | 35 |
| | sp-ReLU-M | Adam | 35 | 100 | 0.25 | 0.0075 | 35 |
| 5-shot | sp-M | Adam | 35 | 100 | 0.25 | 0.0075 | 100 |
| | sp-M$^+$ | SGD+N | 35 | 100 | 0.1 | 0.0075 | 100 |
| | sp-ReLU-M | Adam | 35 | 100 | 0.1 | 0.0075 | 100 |

for the ResNet that we term *large*, with channel sizes $(64, 160, 320, 640)$, and *small*, with channel sizes $(64, 128, 256, 512)$. We also adapted a strategy by [2] where we test on an ensemble of the 3 best models checkpointed while training. The results of all these variants are shown in Table S1.

**Sparse-MAML$^+$.** To generate the underlying mask parameter $m \in \mathbb{R}^N$ in sparse-MAML$^+$ ($N$ being the dimension of the parameter space) we apply an affine transformation to a Gaussian vector $z \in \mathbb{R}^E$ (we set $E = 1600$) with explicitly learnable noise standard deviation $\sigma$:

$$m = A(z \circ \sigma + \mu) + b, \qquad (11)$$

with $A \in \mathbb{R}^{N \times E}, b \in \mathbb{R}^N, \sigma, \mu \in \mathbb{R}^E$ and $z \sim \mathcal{N}(0, I)$. We use this process to generate the gradient mask parameters for convolutional layers only. As with every variant of sparse-MAML studied here, we adjust the meta-parameters $A, b, \mu, \sigma$ using a first-order update and straight-through estimation.

### B.1.2 Additional analyses

Complementing Figure 1, we show in Figure S1 emerging gradient sparsity in batch normalization and bias parameters throughout the network. Interestingly, we observe non-monotonicity in the sparsity levels especially in batch normalization parameters throughout training. This is possible by allowing to change sparsity in both directions by using the straight-through estimator for the binary mask. We find that the bias parameters eventually become entirely frozen (Figure S1 right) irrespective of initialization.

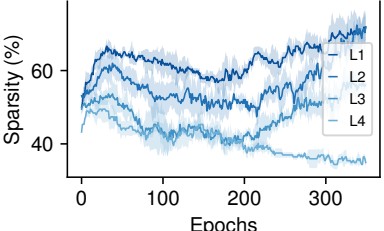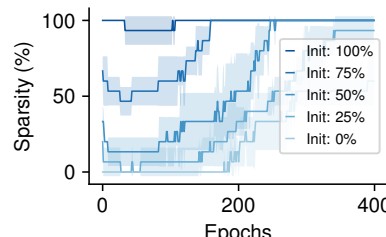

Figure S1: Emergent gradient sparsity in 5-shot 5-way classification of miniImageNet on the standard 4-convolutional-layer neural network, with inner-loop learning rate $0.1$ and $25$ inner-loop steps. Results averaged over 5 seeds $\pm$ std. *Left*: Different final gradient sparsity for batch normalization gain parameters emerges with gradually less sparsity from earlier to deeper layers, all initialized at $50\%$ sparsity. *Right:* Output layer bias parameter sparsity for different initial sparsity levels tend towards $100\%$. Note that deeper layers typically tend towards lower levels of sparsity.

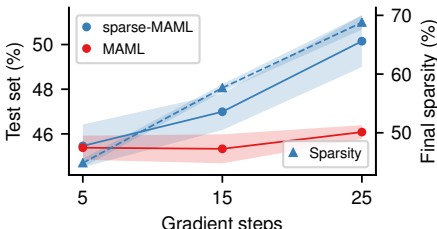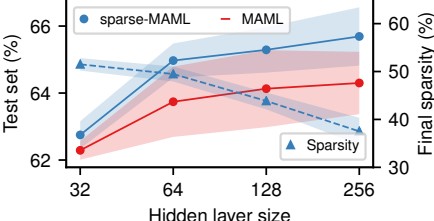

Figure S2: Additional results for 5-way miniImageNet classification, complementing Figure 2. *Left*: In 1-shot learning problems, long inner-loops lead to an increase in generalization performance accompanied by high gradient sparsity levels. By contrast, the performance of standard MAML does not improve with longer inner-loops. *Right:* Gradient sparsity decreases as hidden layer width increases. The inner-loop learning rate was set to $0.1$ for all hidden layer sizes and gradient sparsity is initially $\sim 50\%$. Results are averages over 5 seeds $\pm$ std.

We additionally carry out an analysis of models with varying hidden layer sizes, cf. Figure S2, and find that sparsity is anti-correlated with network width, indicating that the pressure of preventing interference by sparse gradients is reduced in large-capacity models.

We further show the performance of sparse-MAML on models without bias parameters, as these are consistently chosen to be frozen by meta-learning, to verify whether they are useful as task-shared parameters or simply not required at all. We find that performance drops slightly when removing the bias parameters, Table S3, which indicates that sparse-MAML ascribes to these parameters the role of providing useful task-shared bias.

These experiments are complemented by a study of the challenging non-transductive BatchNorm setting. Here, we simply compute batch statistics over the course of meta-train/test training without computing new statistics during meta-train/test testing – we point to ref. [6] for a discussion. Since FOMAML was close to chance-level performance for $35$ inner-loop steps, the results reported for FOMAML are produced with $10$ inner-loop steps and $\alpha = 0.1$. Sparsity emerges again with sparse-MAML, although now without bringing a performance advantage over FOMAML, see Table S3. All

Table S3: Additional 5-way 5-shot miniImageNet few-shot learning experiments investigating the non-transductive batch normalization setting, and an ablation study in which bias parameters (which are consistently frozen by sparse-MAML) are removed from the model.

| Algorithm | Test set acc. (%) |
|---|---|
| sparse-MAML | $67.03^{\pm 0.74}$ |
| sparse-MAML w/o bias parameters | $66.11^{\pm 0.57}$ |
| FOMAML non-transductive BatchNorm | $55.58^{\pm 1.68}$ |
| sparse-MAML non-transductive BatchNorm | $54.75^{\pm 1.17}$ |

Table S4: Two-phase learning experiments: meta-learning a gradient mask after learning the model initialization using standard MAML does not result in improved generalization performance on 5-shot miniImageNet learning.

| Setup | Initial sparsity (%) | Final sparsity (%) | Test set acc. (%) |
|---|---|---|---|
| 1-shot sparse-MAML | 0 | 9 | $46.42^{\pm 0.58}$ |
| 1-shot sparse-MAML | 50 | 45 | $46.42^{\pm 0.27}$ |
| 5-shot sparse-MAML | 0 | 29 | $64.68^{\pm 0.16}$ |
| 5-shot sparse-MAML | 50 | 51 | $64.01^{\pm 0.47}$ |

hyperparameters were kept the same as described in Table S2, except for the change in inner-loop length and $\alpha$ that was needed to stabilize FOMAML.

We present one last few-shot learning study in Table S4, where we test whether meta-learning of the model initialization $\theta$ and the sparsity mask $m$ have to happen jointly, or if an appropriate gradient mask can be found separately after standard MAML training, keeping $\theta$ fixed. We find that this form of meta-learning fails to improve upon standard MAML alone. Thus, the generalization performance improvements brought by sparse-MAML rely on discovering a model initialization that is specialized for sparse learning. These results indicate that it is unlikely that the performance of sparse-MAML can be reached by simply analyzing the MAML solution post-training and heuristically disabling certain weight updates.

## B.2 La-MAML experiments

### B.2.1 Reproducibility

We strictly follow the experimental setup of the original La-MAML study and use the code provided by the authors[2]. The reported results are obtained by scanning three hyperparameters in the same range considered in the original paper, cf. Appendix of ref. [15]. Therefore, we only vary the number of glances within $\{5, 10\}$, the outer-loop mask learning rate $\gamma_m$, and the inner-loop learning rate $\alpha_0$. See Table S6 for the hyperparameters found by our scan. The network used for the MNIST experiments is a 2-hidden-layer neural network with 100 hidden rectified linear units and the number of parameters is 89.610: [(78400, 100), (10000, 100), (1000, 10)] in (no. of weights, no. of biases) format and in input-to-output order. The output layer has a softmax nonlinearity and we use the cross-entropy loss.

---

**Algorithm 1:** One step of sparse-La-MAML

**Require:** Parameters $\theta$, mask parameters $m$, replay buffer $R$, incoming batch of data $\mathcal{B}$, inner-loop learning rate $\alpha_0$, mask learning rate $\gamma_m$, loss $\mathcal{L}$

$\phi \leftarrow \theta$
$g^{\text{in}} \leftarrow 0$
**for** $1 \leq k \leq |\mathcal{B}|$ **do**
$\quad \phi \leftarrow \phi - \alpha_0 \, \mathbb{1}_{m \geq 0} \circ \nabla \mathcal{L}(\phi, \mathcal{B}_k)$
$\quad g^{\text{in}} \leftarrow g^{\text{in}} + \nabla \mathcal{L}(\phi, \mathcal{B}_k)$
$\mathcal{R} \leftarrow$ Sample past data batch from $R$
$m \leftarrow m + \gamma_m \, \alpha_0 \, \nabla \mathcal{L}(\phi, \mathcal{B} \cup \mathcal{R}) \circ g^{\text{in}}$
$\theta \leftarrow \theta - \alpha_0 \, \mathbb{1}_{m \geq 0} \circ \nabla \mathcal{L}(\phi, \mathcal{B} \cup \mathcal{R})$
$R \leftarrow$ Update replay buffer $R$ with $\mathcal{B}$

---

Pseudocode for one complete iteration of sparse-La-MAML can be found in Algorithm 1. The fixed-size replay buffer $R$ is updated stochastically with the reservoir sampling method presented in ref. [49].

Following the original La-MAML experimental setup, we study three supervised continual learning (CL) problems based on MNIST. In MNIST *rotations* (20 tasks, 1000 examples per task), each task is a classification problem where MNIST digits rotated by a fixed task-specific common angle (in $[0, \pi]$) are to be classified. In MNIST *permutations* (20 tasks, 1000 samples per task) and the harder *many permutations* variant (100 tasks, 200 examples per task), a fixed task-specific pixel shuffling order is applied to every MNIST digit instead.

To produce the results in the left panel of Figure 3, we choose the configuration used for MNIST rotations found by our scan (cf. S6) and vary the layer size of the two hidden layers of the fully-

---

[2]https://github.com/montrealrobotics/La-MAML

connected network. For the results shown in Figure 4, we keep the hyperparameters of the original La-MAML paper but iterate over the dataset 10 times (epochs) instead of only once.

### B.2.2 Additional analyses

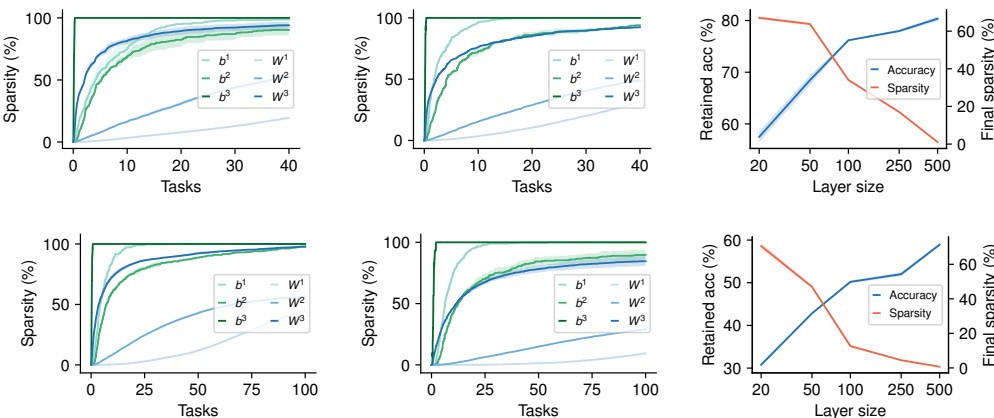

Figure S4: Gradient sparsity when learning on MNIST permutations (upper column) and many permutations (lower column) with La-MAML (first row) and sparse-La-MAML (second and third row). Results averaged over 3 seeds $\pm$ std. *Upper left (original La-MAML algorithm)*: Sparsity emerges across the three-layer network and monotonically increases with the number of tasks and with depth for both weight ($W_1, W_2, W_3$) and bias parameters ($b_1, b_2, b_3$). *Center (sparse-La-MAML)*: A similar behavior is observed when replacing meta-learned learning rates by meta-learned binary gradient masks. *Right (sparsity/accuracy vs. layer size):* Overall sparsity of sparse-La-MAML decreases with increased network capacity accompanied with higher retained accuracy (RA). Network capacity is varied by changing the number of neurons in the two hidden layers simultaneously.

For completeness, we visualize the patterns of gradient sparsity that emerge when learning continually with La-MAML and sparse-La-MAML on the MNIST permutations and many permutations CL problems, see Figure S4. The findings reported in the main text translate to these two datasets, and the two variants of La-MAML again behave in a qualitatively similar way.

In our experiments, we observe that the inner-loop learning rate $\alpha_0$ has a strong effect on gradient sparsity. This is depicted in Figure S3 where the final gradient sparsity level for sparse-La-MAML trained on MNIST permutations is shown, together with retained accuracy. We find that while sparsity and accuracy are jointly maximized for lower inner-loop learning rate $\alpha_0$, high retained accuracies can still be achieved when increasing the learning rate $\alpha_0$, up to a point where accuracy eventually drops.

Table S5: Sparsity (%) of La-MAML and sparse-La-MAML after learning on one of the three MNIST continual learning problems rotations, permutations and many permutations. Hyperparameters were tuned for best retained accuracy, not sparsity. We split between weight and bias parameters (weights followed by bias) when presenting per-layer average levels of sparsity within layers (ordered from network input to output). Structured gradient sparsity emerges, with lower-levels of sparsity for lower-level features closer to the input. Bias parameters tend to be close to frozen in almost all cases.

| Dataset | Algorithm | Average within layers (%) | Average (%) |
|---------|-----------|---------------------------|-------------|
| Rotations | La-MAML | [16, 45, 96], [95, 94, 100] | 20.47 |
| | sparse-La-MAML | [5, 19, 80], [67, 54, 100] | 7.13 |
| Permutations | La-MAML | [20, 53, 97], [99, 93, 100] | 24.81 |
| | sparse-La-MAML | [8, 29, 78], [100, 87, 100] | 16.17 |
| Many permutations | La-MAML | [45, 56, 98], [100, 98, 100] | 46.53 |
| | sparse-La-MAML | [10, 29, 87], [100, 93, 100] | 13.08 |

Table S6: Hyperparameter settings for the reported La-MAML and sparse-La-MAML results.

| Dataset | | Algorithm | $\alpha_0$ | $\gamma_m$ | $K$ / Glances |
|---|---|---|---|---|---|
| MNIST | Rotations | LaM | 0.15 | 0.3 | 5 |
| | | sp-LaM | 0.15 | 1.7 | 5 |
| | Permutations | LaM | 0.15 | 0.3 | 5 |
| | | sp-LaM | 0.1 | 1.7 | 10 |
| | Many | LaM | 0.1 | 0.3 | 10 |
| | | sp-LaM | 0.05 | 0.75 | 10 |

Table S7: Final full CIFAR-10 test-set classification accuracy, continually-learned in a class-incremental, streaming fashion, in 5 tasks comprising 2 classes each. Each data point is seen only once. We compare sparse-La-MAML (sp-LaM; binary gradient masks, straight-through update), standard La-MAML (LaM; rectified learning rates, meta-learned without straight-through update), experience replay, gradient episodic memory (GEM) and meta-experience replay (MER), for two different replay buffer sizes. Results are averages over 5 seeds $\pm$ std.

| Total memory size | Algorithm | Final acc. (%) |
|---|---|---|
| 200 | Experience replay | $19.75^{\pm1.23}$ |
| | MER | $25.11^{\pm1.77}$ |
| | GEM | $25.14^{\pm0.67}$ |
| | LaM | $22.08^{\pm1.83}$ |
| | sp-LaM | $27.85^{\pm0.69}$ |
| 1000 | Experience replay | $29.12^{\pm2.41}$ |
| | MER | $34.66^{\pm1.38}$ |
| | GEM | $31.55^{\pm0.81}$ |
| | LaM | $36.24^{\pm0.91}$ |
| | sp-LaM | $37.70^{\pm0.80}$ |

**Streaming Split-CIFAR-10 experiments.** Finally, we complement our continual learning investigation of gradient sparsity in La-MAML with results on a streaming Split-CIFAR-10 class-incremental learning problem. In this problem, the CIFAR-10 dataset is split into 5 tasks of 2 classes each, and each data point is processed online only once. We use a 4-convolutional-layer neural network, the same used in the original La-MAML paper [15]. This is a challenging setting where experience replay (ER) remains a strong baseline [1]. We produced this baseline for our architecture, and compared sparse-La-MAML to it, performing for both methods a hyperparameter scan over replay batch size, the number of gradient updates per incoming batch, and learning rates. We also compare to GEM (while optimizing the

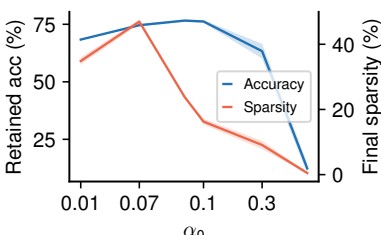

Figure S3: Retained accuracy (RA) and final gradient sparsity levels (in %) for sparse-La-MAML applied to MNIST permutations, for different settings of the inner-loop learning rate $\alpha_0$.

following hyperparameters: batch sizes, number of gradient updates per batch, gradient clipping norm, the strength with which the memory constraint is enforced) and MER (scanning over batch sizes, regularization strength, gradient clipping norm, and learning rates).

For both small (20 examples per class) and large (100 examples per class) replay buffer sizes, sparse-La-MAML consistently outperforms ER, cf. Table S7, as well as MER and GEM.

We observe a qualitatively distinct gradient sparsity pattern emerge in this setting, compared to our MNIST experiments. Here, sparse-La-MAML leads to a large fraction of frozen weights in the final fully-connected layer ($57.5 \pm 0.866\%$ for the small replay buffer case, and $64.75 \pm 0.8292\%$ for the

large replay buffer), which increase as more classes are learned, and significantly lower values of gradient sparsity for the remaining parameters ($1.6\pm0.68\%$ overall sparsity for the small replay buffer case, and $1.0\pm0.70\%$ for the large replay buffer). Our results confirm that the strong performance of La-MAML translates to a more challenging class-incremental continual learning problem, and reveal that meta-learning finds a solution with large gradient sparsity in the final output layer.

We further compare to the original La-MAML implementation provided in [16] for the MNIST experiments, which uses rectified learning rates (Eq. 7) but not our straight-through update. As discussed in the main text, this leads to dead parameter updates that can never recover once the learning rate goes below zero. We find that this variant of La-MAML leads to very high levels of gradient sparsity in the entire model ($96.3\pm1.5\%$ when using small replay buffers, and $34.0\pm1.39\%$ when using large replay buffers) but a performance hit, highlighting the importance of fine-tuning gradient masks without aggressively shutting off learning.

We found the following hyperparameters to work the best for each particular method:

- *La-MAML*. Batch size and replay batch size: 10; number of gradient steps per data point: 2; For memory size 200, 1000 we used $\alpha_{\text{init}} = 0.005, 0.01$ and $\gamma = 0.1, 0.01$, resp.

- *GEM*. Number of steps per data point: 2; memory strength: 0.5; 500 samples per task. For memory sizes of 200 and 1000, we use batch sizes of 20 and 5, resp.

- *MER*. Batch size and replay batch size: 10, $\beta = 0.1$. $\gamma = 0.05$, $\gamma = 0.08$ for memory sizes of 200 and 1000, resp.

- *ER*. Batch size of 10 for both memory sizes. Gradient steps per data point: $2, 4$, $\gamma$: 0.001, 0.01; replay batch size: $20, 10$, for memory sizes of 200 and 1000 resp.

## B.3 C-MAML experiments

We provide the full performance overview of the different C-MAML variants studied in the main text together with related work in Table S8.

### B.3.1 Reproducibility and ablation study

In order to obtain the reported results, we use the code base[3] that accompanies ref. [7]. We do not alter the architecture of the 4-convolutional-layer neural network (64 hidden units) used in the original C-MAML study. All our results are based on the best performing, non-ablated version of the C-MAML algorithm, termed C-MAML+UM+PAP in the original paper [7]. Furthermore, we do not change the provided hyperparameters, and only tune the inner-loop and mask learning rates $\alpha_0$ and $\gamma_m$ (resp.) for our sparse-C-MAML and sparse-ReLU-C-MAML algorithm variants. For sparse-C-MAML in the $p = 0.98$ setup, we initialized the mask parameters with the Kaiming initialization leading to an initial sparsity of $50\%$. For all sparse-ReLU-C-MAML runs, we initialized the mask parameters with a uniform initialization over the range $[0.005, 0.1]$.

---

[3]`https://github.com/ElementAI/osaka`

Table S8: Cumulative online accuracy on the Omniglot-MNIST-FashionMNIST online learning benchmark as well as the accuracy on the single tasks. Tasks switch with probability $1 - p$. Results from previous work taken from ref. [7]. Mean $\pm$ std. over 5 seeds.

| METHOD | $p = 0.98$ | | | | $p = 0.90$ | | | |
|---|---|---|---|---|---|---|---|---|
| | TOTAL | OMNIGLOT | MNIST | FASHION | TOTAL | OMNIGLOT | MNIST | FASHION |
| ONLINE ADAM | $73.9^{\pm2.2}$ | $81.7^{\pm2.3}$ | $70.0^{\pm3.6}$ | $62.3^{\pm2.5}$ | $23.8^{\pm1.2}$ | $26.6^{\pm2.0}$ | $20.0^{\pm1.4}$ | $22.1^{\pm1.3}$ |
| FINE TUNING | $72.7^{\pm1.7}$ | $80.8^{\pm2.0}$ | $68.7^{\pm2.8}$ | $59.6^{\pm3.1}$ | $22.1^{\pm1.1}$ | $25.5^{\pm1.5}$ | $18.1^{\pm1.9}$ | $19.2^{\pm1.6}$ |
| MAML [11] | $84.5^{\pm1.7}$ | $97.3^{\pm0.3}$ | $80.4^{\pm0.3}$ | $63.5^{\pm0.3}$ | $75.5^{\pm0.7}$ | $88.8^{\pm0.4}$ | $68.1^{\pm0.5}$ | $56.2^{\pm0.4}$ |
| ANIL [44] | $75.3^{\pm2.0}$ | $95.1^{\pm0.6}$ | $58.7^{\pm2.9}$ | $49.7^{\pm0.3}$ | $69.1^{\pm0.8}$ | $88.3^{\pm0.5}$ | $52.4^{\pm0.6}$ | $47.6^{\pm0.9}$ |
| BGD [60] | $87.8^{\pm1.3}$ | $95.1^{\pm0.5}$ | $86.9^{\pm1.1}$ | $74.4^{\pm1.1}$ | $63.4^{\pm0.9}$ | $72.8^{\pm1.2}$ | $55.9^{\pm2.2}$ | $51.7^{\pm1.3}$ |
| METACOG [19] | $88.0^{\pm1.0}$ | $95.2^{\pm0.5}$ | $87.1^{\pm1.5}$ | $74.3^{\pm1.5}$ | $63.6^{\pm0.9}$ | $73.5^{\pm1.3}$ | $55.9^{\pm1.8}$ | $51.7^{\pm1.4}$ |
| METABGD [19] | $91.1^{\pm2.6}$ | $96.8^{\pm1.5}$ | $92.5^{\pm1.9}$ | $77.8^{\pm3.8}$ | $74.8^{\pm1.1}$ | $83.1^{\pm1.0}$ | $71.7^{\pm1.5}$ | $61.5^{\pm1.2}$ |
| C-MAML | $92.8^{\pm0.6}$ | $97.8^{\pm0.2}$ | $93.9^{\pm0.8}$ | $79.9^{\pm0.7}$ | $83.3^{\pm0.4}$ | $89.0^{\pm0.5}$ | $84.5^{\pm0.7}$ | $71.1^{\pm0.7}$ |
| SPARSE-C-MAML | $94.2^{\pm0.4}$ | $97.3^{\pm0.1}$ | $93.4^{\pm0.4}$ | $86.3^{\pm0.3}$ | $86.3^{\pm0.4}$ | $89.3^{\pm0.5}$ | $87.7^{\pm0.4}$ | $77.4^{\pm0.5}$ |
| SPARSE-RELU-C-MAML | $93.5^{\pm0.5}$ | $97.16^{\pm0.2}$ | $97.2^{\pm0.2}$ | $94.1^{\pm0.4}$ | $84.7^{\pm1.3}$ | $89.3^{\pm0.2}$ | $87.5^{\pm0.5}$ | $78.3^{\pm0.2}$ |

---

**Algorithm 2:** One step of sparse-C-MAML

---

**Require:** Current parameters $\phi$, meta-parameters $\theta$, mask parameters $m$,
        replay buffer $R$, incoming batch of data $\mathcal{B}$, inner-loop learning rate
        $\alpha_0$, mask learning rate $\gamma_m$, loss function $\mathcal{L}$, learning rate
        adaptation function $g$

**if not** *task change detected* **then**
    $\phi \leftarrow \phi - \alpha_0 \, \mathbb{1}_{m \geq 0} \circ \nabla \mathcal{L}(\phi, \mathcal{B})$
    $R \leftarrow R \cup \mathcal{B}$   `// update replay buffer with current data`

**else**
    $\mathcal{R}^{\text{t}} \leftarrow$ sample batch of training data from $R$
    $\phi \leftarrow \theta - \alpha_0 \, \mathbb{1}_{m \geq 0} \circ \nabla \mathcal{L}(\theta, \mathcal{R}^{\text{t}})$
    $\mathcal{R}^{\text{v}} \leftarrow$ sample batch of validation data from $R$
    $\eta \leftarrow g(\mathcal{L}(\phi, \mathcal{R}^{\text{v}}))$   `// adapt learning rate`
    $\theta \leftarrow \theta - \eta \nabla \mathcal{L}(\phi, \mathcal{R}^{\text{v}})$
    $R \leftarrow \{\}$   `// reset replay buffer`
    $\phi \leftarrow \theta - \alpha_0 \, \mathbb{1}_{m \geq 0} \circ \nabla \mathcal{L}(\theta, \mathcal{B})$

---

We provide pseudocode for one iteration of sparse-C-MAML in Algorithm 2. Following Caccia et al. [7] and do not use any pretraining; the base parameters $\theta$ and the current parameters used for prediction $\phi$ are initialized randomly and equal to one another. The replay buffer $R$ is also initially empty. For details on the task change detection function and outer-loop learning rate adaptation function we refer to the original C-MAML study [7].

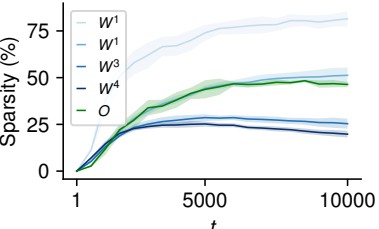

Figure S5: Gradient sparsity when learning online with sparse-C-MAML. Gradient sparsity decreases with depth and rises again for the output layer.

**Omniglot-MNIST-FashionMNIST setup.** The Omniglot-MNIST-FashionMNIST benchmark studied here was introduced in the original C-MAML paper [7]; we do not modify the experimental setup. In this online learning problem, at every time step $t$ the task changes with probability $1 - p$. Each task is a $\mathcal{K}$-shot, 10-way classification problem. Tasks are created by sampling 10 classes uniformly (for Omniglot; MNIST and FashionMNIST are by default 10-way problems) and then sampling $\mathcal{K}$ examples for each of the selected 10 classes.

**Gradient masking ablation study.** In order to verify the advantage of gradient masking, we also compare to an ablated version of C-MAML (called C-MAML-fixed) which does not feature any meta-learned learning rate parameters, setting the inner-loop learning rate to a fixed hyperparameter value (we note that the original C-MAML algorithm included a small set of meta-learned learning rates that were shared for large parameter groups and which were not restricted to be non-negative). Results are shown in Table S9. In the Omniglot-MNIST-FashionMNIST experiment, the performance of C-MAML is matched by C-MAML-fixed.

In all of our experiments, we observed sparsity emerging and higher overall average accuracies for sparse-C-MAML compared to C-MAML-fixed and C-MAML. Note that the only difference between sparse-C-MAML and C-MAML-fixed is the ability to stop learning some of the parameters.

## C   Brief discussion on meta-learning-based approaches to continual learning

The surge of meta-learning in continual learning can be explained by its ability to automatically discover the inductive biases that are appropriate for learning without forgetting. Previous works hypothesize that a particular inductive bias will mitigate catastrophic forgetting, e.g. keeping parameters from diverging too much from previous versions [26], and then develop a solution around that. Contrarily, meta-learning based approaches will learn inductive biases that are conducive for learning without interference in a data-driven way. For example, in ref. [23] sparsity emerges in the learned

representations, a characteristic that has long been hypothesized as desirable in continual learning [13].

Regularization-based methods are notoriously incapable of working in more realistic settings, such as those considered in our work, mostly because they are not equipped with a mechanism to perform cross-task discrimination or to recalibrate themselves on past tasks after some interference has occurred. The same applies for dynamic architectures, which rely on task labels. This reliance can be bypassed with a task-inference module, which may however suffer from some forgetting itself.

Rehearsal-based methods do not suffer from the aforementioned weaknesses. Nevertheless, they scale poorly due to their reliance on always approximating an i.i.d. distribution at every update. The total runtime of these methods scales quadratically with the number of tasks.

The ambitious goal of meta-learning inductive biases that benefit continual learning directly from data may come at the cost of decreasing sample efficiency and increasing compute requirements. However, the latter is potentially offset by reducing the number of hyperparameter-search trials [37].

## D  Resources

**Compute.**   We used 24 (3x8 servers) NVIDIA GeForce 2080 Ti GPUs for our experiments and conducted experiments and hyperparameter scans for approximately one month in order to obtain the reported results.

Table S9: Task-averaged cumulative online accuracy of C-MAML, C-MAML-fixed and sparse-C-MAML and the hyperparameters that lead to the result.

|  | Method | Accuracy (%) | $\alpha_0$ | $\gamma_m$ |
|---|---|---|---|---|
| $p = 0.9$ | C-MAML | $83.3^{\pm 0.4}$ | 0.1 | 0.001 |
|  | C-MAML-fixed | $85.3^{\pm 0.5}$ | 0.3 | - |
|  | sparse-C-MAML | $86.3^{\pm 0.4}$ | 0.3 | 0.003 |
|  | sparse-ReLU-C-MAML | $86.1^{\pm 0.2}$ | - | 0.01 |
| $p = 0.98$ | C-MAML | $92.8^{\pm 0.6}$ | 0.1 | 0.005 |
|  | C-MAML-fixed | $92.0^{\pm 0.1}$ | 0.1 | - |
|  | sparse-C-MAML | $94.2^{\pm 0.4}$ | 0.3 | 0.01 |
|  | sparse-ReLU-C-MAML | $93.5^{\pm 0.4}$ | - | 0.01 |

**Software, libraries and licensing information.**   The results reported in this paper were produced with open source, free software whenever possible. We developed custom code in Python using the PyTorch (BSD-style license) [43] and NumPy (BSD-style license) [17] libraries; few-shot learning dataset splits and meta-gradient computations further relied on the Torchmeta library (MIT license) version 1.6 [10]. Our extensions of the La-MAML (Apache-2.0 license) and C-MAML (unknown license; permission to extend granted by the authors) algorithms were built directly on top of the code distributed by the authors. All plots were generated using matplotlib (BSD-style license) [20]. Our computers run Ubuntu Linux.

We investigated our learning algorithms on the public domain datasets MNIST (GNU GPL v3.0) [30], FashionMNIST (MIT license) [59], Omniglot [29] (MIT license), miniImageNet [47] (custom MIT/ImageNet license), CIFAR-10 (MIT license) [28], CUB (custom license) [57], tieredImageNet (custom ImageNet license) [48] and Cars (custom license) [27].

## E  PyTorch code snippet

In all of our experiment we backpropagate through binary or ReLU masks using the straight-through estimator. For illustrative reasons, we provide a Python code snippet showing how to use either the ReLU straight-through or the binary mask e.g. inside an inner loop of MAML:

Listing 1: Backpropagate through binary or ReLU mask

```python
import torch
class Binary(torch.autograd.Function):
    def __init__(self):
        super(Binary, self).__init__()
    @staticmethod
    def forward(ctx, input):
        return torch.sign(input)
    @staticmethod
    def backward(ctx, grad_output):
        return grad_output

class ReLUThrough(torch.autograd.Function):
    def __init__(self):
        super(ReLUThrough, self).__init__()
    @staticmethod
    def forward(ctx, input):
        return torch.relu(input)
    @staticmethod
    def backward(ctx, grad_output):
        return grad_output

def training():
    ...
    # Inside an inner loop
    grads = torch.autograd.grad(loss, weights)
    if ReLUThroughMask:
        params = params - ReLUThrough.apply(m)*grads
    elif BinaryMask:
        # alpha is the inner loop learning rate and a hyperparameter
        params = params - alpha*0.5*(Binary.apply(m)+1)*grads
```