# OpenReview forum: "Learning where to learn: Gradient sparsity in meta and continual learning"
_NeurIPS.cc/2021/Conference — NeurIPS 2021 Poster_

### Official Review · Reviewer_WL2f · 2021-07-11

**Rating:** 7
**Confidence:** 5

**Summary:**

- This paper introduces a sparsity regularizer for gradient-based meta learners. A binary mask for each layer is used to decide whether a parameter should be adapted or not during the inner loop phase.
- The authors provide an analysis of sparse-MAML on standard few-shot learning   (FSL) datasets in terms of performance (accuracy) and gradient sparsity. They also verify the performance of sparse-MAML in the cross-domain setting comparing it with BOIL and ANIL, other work manually freezing the head or the body of the classifier, rather than learning what to freeze/adapt as sparse-MAML does. The second part of the paper revolves around look-ahead MAML, an algorithm for continual learning investigating the gradient sparsity caused by meta-learning the inner learning rate (with rectifier, non-negative LR, or binary mask). Finally, experiments on online learning meta/multitask learning are provided.


**Limitations And Societal Impact:**

Some limitations are discussed. No potential negative impact in my opinion.

**Main Review:**

- The paper is clearly written and well explained. Every concept is introduced with particular attention and the rationale of each section is explained step by step making.
- The main limitation of this approach is the necessity of having a binary mask to cover all parameters. It is definitely challenging (interesting) scaling the proposed method to other architectures. I believe this is the reason why the authors do not provide experiments with other architectures but always use the same conv4 vgg-like model (also for easier comparison with existing approaches). Authors leave this issue for future work.
- **Questions/considerations on experiments**
    1. It would have been great to have a comparison with other sparsity regularizers. A simple differentiable L1/L0 constraint on the weights could be an interesting experiment, although much more difficult to train and requiring to tune a sparsity coefficient hyperparameter.
    2. A comparison with Shrinkage meta-learning [6] algorithm is also missing.
    3. I guess my main question is: "Is sparsity really necessary for meta-learning?" I'm not sure the paper really wants to answer this question, but I find it interesting. Since the proposed approach claim sparsity to be important for generalization in meta/continual learning it would have been nice to see experiments with masks being free to vary, rather than being constrained to be binary. This would be important to reinforce the claim about the importance of sparsity.  I believe that would correspond to something similar to MT-Net [20] or warpgrad[9].
    4. It is important to notice that when you add per-parameters LR or binary masks you double the number of parameters. Considering this point, how is your comparison fair wrt MAML and other methods?
    5. I see in the Supp. Mat. that training is performed following the transductive usage of BatchNorm. I would also include experiments with non-transductive settings to disentangle the effect of bn statistics.
- Pros
    - Code, ablation studies, and information for reproducibility are present in the supplementary material


——————————————-
After the rebuttal and discussions, I decided to increase my score from 5 to 7 and recommend acceptance.
The paper is an important empirical contribution to the fields of Meta and Continual Learning that might be useful for some researchers.
The authors have agreed to include new results and important changes/clarifications to the final version of the paper.
If accepted, these changes should be checked by the AC and reviewers when the camera-ready is uploaded.


**Time Spent Reviewing:**

4.5

---

> ### Author Response · Authors · 2021-08-10
> **Reply to Reviewer WL2f**
>
> Thank you for your thorough review and interesting comments. We reply to your concerns and suggestions point-by-point below:
>
> ---
> > The main limitation of this approach is the necessity of having a binary mask to cover all parameters. It is definitely challenging (interesting) scaling the proposed method to other architectures. I believe this is the reason why the authors do not provide experiments with other architectures but always use the same conv4 vgg-like model (also for easier comparison with existing approaches)
>
> It is computationally intensive and difficult to apply meta-learning methods based on MAML (including ours) to more complex architectures such as ResNets. We definitely agree that scaling up MAML is an important direction of future work.
>
> Still, we would like to point out that among the cited variants of MAML we compare and refer to, our method is by far the lightest of all. With our sparse-MAML, after meta-learning, only one extra bit needs to be stored per parameter (the other methods require floats and/or non-diagonal modulation matrices, the latter scaling quadratically with the number of parameters). Moreover, maintaining the auxiliary mask parameter float during meta-learning is similar to what is required by many widely-adopted optimizers such as ADAM, which also require storing one or more per-parameter auxiliary variables during learning.
>
> ---
> > It would have been great to have a comparison with other sparsity regularizers. A simple differentiable L1/L0 constraint on the weights could be an interesting experiment, although much more difficult to train and requiring to tune a sparsity coefficient hyperparameter.
>
> Promoting weight sparsity (as opposed to gradient sparsity) is an interesting and well-studied approach to improve generalization performance. However, we feel that it is outside the scope of our work, which focuses on gradient modulation, and in particular, in sparse gradients. Regarding explicit gradient regularization, the reason why we did not pursue this direction (e.g. an L1 penalty on the gradients) is that this would require computing expensive second-order derivatives; one of our main aims was to focus on inexpensive first-order meta-learning methods. Moreover, one of our main findings is that gradient sparsity emerges in meta-learning when optimizing for generalization to a heldout dataset, and without any explicit regularization towards it. Following your comment, we added this point to the discussion section, noting that a comparison to gradient regularizers could be an interesting direction for future study.
>
> ---
> > A comparison with Shrinkage meta-learning [6] algorithm is also missing.
>
> We now added the results from Table 10 of [6] to our Table 1 and expanded our discussion of this method in section 3.4. Briefly, it also discovers a similar pattern of frozen task-shared features, but on miniImageNet it does not improve learning performance as much as sparse-MAML. We note that as the authors of [6] do not provide accompanying code it is difficult to carry out a more extensive detailed comparison.
>
> ---
> > I guess my main question is: "Is sparsity really necessary for meta-learning?" I'm not sure the paper really wants to answer this question, but I find it interesting. Since the proposed approach claim sparsity to be important for generalization in meta/continual learning it would have been nice to see experiments with masks being free to vary, rather than being constrained to be binary. This would be important to reinforce the claim about the importance of sparsity. I believe that would correspond to something similar to MT-Net [20] or warpgrad[9].
>
> This control corresponds to the existing meta-SGD (in few-shot learning) and La-MAML (in continual learning) baselines that we compare to. Note that we also implemented our own sparse-ReLU-MAML control which uses our first-order straight-through estimation, and for which we ran our own extensive hyperparameter scans, to make an even finer comparison. Interestingly, in such methods, gradient sparsity also emerges (cf. Table S1; this result also holds for La-MAML, Table 3). Thus, gradient sparsity seems to be a robust emergent phenomenon in gradient modulation meta-learning methods, which results in improved learning performance.
> Using the extra page and following your comment we now present Table S1 in section 3.4 of the main text.
>
> ---
> > It is important to notice that when you add per-parameters LR or binary masks you double the number of parameters. Considering this point, how is your comparison fair wrt MAML and other methods?
>
> Our method falls into a class of extensions of MAML which learn how to modulate gradients, such as Meta-SGD, Meta-Curvature, MT-net, and warpgrad. Other related methods, such as meta-shrinkage, learn regularizers. All of these alternatives to sparse-MAML increase the number of meta-parameters over vanilla MAML, aiming to improve it. Thus, the comparison to them is fair, with vanilla MAML serving as a reference baseline. We would like to point out that Sparse-MAML is in fact comparable in cost to meta-SGD (which in turn uses the cheapest form of gradient modulation among existing methods) during meta-learning, and less expensive during deployment, as only one bit per parameter needs to be saved.
>
> ---
> > I see in the Supp. Mat. that training is performed following the transductive usage of BatchNorm. I would also include experiments with non-transductive settings to disentangle the effect of bn statistics.
>
> For completeness and following your suggestion we will include an experiment in the final version of the SM in the non-transductive BatchNorm setting.
>
> ---
> Given the changes and clarifications detailed above, we kindly ask you to reevaluate our paper and consider recommending it for acceptance. If you have any further concerns or comments please do not hesitate in contacting us during the rolling discussion period.

---

> > ### Author Response · Authors · 2021-08-20
> > **Follow-up on scalability question**
> >
> > Following your question on scalability to more complex architectures we pursued additional few-shot learning experiments on a ResNet-12 model, which have now concluded. We would like to draw your attention to our joint reply to all reviewers containing these new results.

---

> > > ### Comment · Reviewer_WL2f · 2021-08-24
> > > **Reponse to rebuttal**
> > >
> > > Dear authors,
> > > Thank you for your clarifications and sorry for my delayed response!
> > >
> > > I’ve read again the paper in a different light thanks to the points raised by other reviewers and your detailed answers. In particular, I agree with rev C1hv on the fact that the narrative of the paper should be improved and that experiments' presentation often lacks a clear justification on their importance and on the insights we can collect from the results. The impression is to be in front of a list of things that you tried and the results you obtained. I agree with rev C1hv that there should be a more natural and complete presentation. It is crucial to add more justifications and put both experiments and results in perspective by always remembering the bigger picture in which they are living. Overall, I think that this is fixable and that the changes you propose in your answers are already pointing in the direction of improving the narrative and the motivation of the paper.
> > >
> > > I’ve appreciated your effort to provide the new results on resnet12 architecture, which support your previous findings and in my opinion make the paper stronger.
> > >
> > > Ultimately, I think that this is a solid empirical paper, which results and findings will be useful for the meta-learning community. I will further discuss the paper with the other reviewers and AC before changing my rate.

---

> > > > ### Author Response · Authors · 2021-08-31
> > > > **Thank you for the updated review**
> > > >
> > > > Dear reviewer WL2f,
> > > >
> > > > Many thanks for the lively discussion and enthusiastic support of our work. The proposed changes will appear in the final version of the paper.

---

### Official Review · Reviewer_ah52 · 2021-07-13

**Rating:** 6
**Confidence:** 3

**Summary:**

This paper proposes a method to selectively update parameters of a model in the paradigm of meta learning, which introduces  sparsity into the updates and can find a better initialization for learning further tasks. It can be applied in few-shot learning, domain adaptation, continual learning, and online learning.



**Limitations And Societal Impact:**


The limitation of experiments are stated above. No societal impact in my understanding.

**Main Review:**

The paper is well written and the proposed method is clearly justified. Comparing with related methods, the difference is not significant but there are some interesting experimental results showing the sparsity is highly related to the model structure and brings some improvements based on MAML.

The authors try to exhibit the potential of this method in different learning paradigms, however, the page limitation wouldn't allow adequate experiments and analysis in all of these paradigms, which makes some claims are not convincing enough.
Eq.5 shows that only parameters with non-negative product of  \nabla L^{out} and \sum \nabla L^{in} will be updated. It is even more restrictive than GEM in continual learning which just requires the inner product of the two gradient vectors is non-negative. Such restriction  is likely to reduce the plasticity in the model. However, the experiments in continual learning only include Domain-IL tasks. How the proposed method performs in Class-IL tasks in not clear.  And it is strange that this method has higher BTI than baseline and MER in Ratation MNIST tasks, which is not explained or  analyzed.

Some minor issues:
1. How does the 100% initial sparsity work? Does it mean there is no update at all in the beginning?
2. In Tab.3, bias parameters tend to be frozen? Why is that? Would it be better without the bias parameters?


**Time Spent Reviewing:**

3 hours

---

> ### Author Response · Authors · 2021-08-10
> **Reply to Reviewer ah52**
>
> Thank you for the useful criticism and questions. We reply below to each point raised in your review:
>
> ---
> > Eq.5 shows that only parameters with non-negative product of \nabla L^{out} and \sum \nabla L^{in} will be updated. It is even more restrictive than GEM in continual learning which just requires the inner product of the two gradient vectors is non-negative. Such restriction is likely to reduce the plasticity in the model.
>
> We would like to note that GEM strictly enforces the gradient inner product non-negativity constraint at every update step for all data points in the buffer, while our method slowly determines for which coordinates plasticity should be blocked by accumulating an elementwise product over many steps. As a result, it usually requires multiple update steps to disable the plasticity of a specific parameter. Due to this difference it seems difficult to compare the two methods directly.
>
> We further note that other studies have found that GEM can be underperforming when compared to simpler baseline methods such as experience replay [Chaudhry19]. Our experiments suggest that sparse-La-MAML is able to find a better stability-plasticity tradeoff, despite featuring an explicit plasticity block in a (dynamic) parameter subset.
>
> [Chaudhry19] Chaudhry, Arslan, et al. (2019). On tiny episodic memories in continual learning. arXiv preprint arXiv:1902.10486
>
> ---
> > The experiments in continual learning only include Domain-IL tasks. How the proposed method performs in Class-IL tasks in not clear.
>
> Following your comment we ran additional experiments evaluating sparse-La-MAML using a convolutional neural network (the same used in the La-MAML paper [10]) on a Split-CIFAR-10 class-incremental learning problem (5 tasks of 2 classes each, each data point seen online once). This is a challenging setting where experience replay (ER) remains a strong baseline (cf. [Aljundi19]). We produced this baseline for our architecture, and compared sparse-La-MAML to it, performing for both methods a hyperparameter scan over replay batch size, the number of gradient updates per incoming batch, and learning rates. For both small (20 examples per class) and large (100 examples per class) replay buffer sizes, our method consistently outperforms ER, cf. table below. We observe a qualitatively distinct gradient sparsity pattern emerge in this setting, with a large fraction of frozen weights in the final fully-connected layer, which increase as more classes are learned, and significantly lower values of gradient sparsity for the remaining parameters. These results are now part of the main text. For the final version of the paper we will complete our results with GEM, MER, and standard La-MAML.
>
> | Setup (replay buffer size) | Final acc. (%)  |
> | :------------- |:-------------:|
> | Experience replay (20)   | 19.75 +/- 1.23   |
> | sparse-La-MAML (20)   | 27.85 +/- 0.69  |
> | Experience replay (100) | 29.12 +/- 2.41   |
> | sparse-La-MAML (100)    | 37.70 +/- 0.80 |
>
> ---
> > It is strange that this method has higher BTI than baseline and MER in Rotation MNIST tasks, which is not explained or analyzed.
>
> The lower baseline BTI (smaller average change in absolute accuracy) can be explained by the lower overall accuracies achieved by it; MER is a competitive alternative method, which appears to achieve a retained accuracy comparable to ours, by reducing the level of forgetting. We added this point to the text.
>
> ---
> > How does the 100% initial sparsity work? Does it mean there is no update at all in the beginning?
>
> A somewhat counter-intuitive feature of our first-order mask update (eq. 5) is that it still updates the mask, even when it is initialized at zero (note that the change is based on the unmasked gradients).
>
> ---
> > In Tab.3, bias parameters tend to be frozen? Why is that? Would it be better without the bias parameters?
>
> It is unclear if this implies whether biases can be removed altogether from the model, or if they play a useful role as task-shared (frozen) parameters. We will investigate this issue for the final version of the paper.
>
> ---
> We would be very grateful for a reassessment of our work, given the clarifications above and our new continual learning results. We believe that these results have improved our paper, complementing an already strong few-shot learning section, which features extensive analyses, custom controls, and cross-domain experiments that go beyond the standard evaluation setting. We remain open during the upcoming rolling discussion period to any further questions or suggestions you may have.

---

> > ### Comment · Reviewer_ah52 · 2021-09-01
> > **Thanks for your feedback**
> >
> > I think authors clarified most of my concerns and I raised my score to 6.

---

> > > ### Author Response · Authors · 2021-09-02
> > > **Thanks for the review update**
> > >
> > > Dear reviewer ah52,
> > >
> > > We are happy to hear that your main concerns have been answered. Many thanks for the reevaluation of our work.

---

### Official Review · Reviewer_giJs · 2021-07-14

**Rating:** 7
**Confidence:** 4

**Summary:**

This paper proposes learning to where to learn to decide which neural network weights to change.  The proposed sparse-MAML introduces an adjustable binary mask that masked gradients on a per-parameter basis, therefore determining which parameters are allowed to change.  The sparse-MAML is also used for continual learning, which could improve generalization and reduce forgetting. The experiment results suggest that learning by sparse gradient descent can learn a powerful inductive bias for meta-learning systems.

**Ethics Review Area:**

["I don’t know"]

**Limitations And Societal Impact:**

(1) Perhaps missing some additional details that would help the reader better understand why and how the method works. See the clarity.

(2)  As the limitations of the paper say,  this paper introduced two important approximations, which may result in some small gaps from the real objective.

(3) One small question: Is the sparse-MAML based on transductive batch normalization? It needs to be clarified in the paper.


**Main Review:**

Quality: The paper is generally well written and easy to follow., and the experiments are thorough and well-executed.  There are extensive experiments demonstrating the empirical effectiveness of the proposed approach.  The proposed learning where to learn decide which weights to change. The selective sparsity achieves better generalization in a range of few-shot and continual learning problems. The versatility experiment is a bonus.

Clarity: Most of my concerns about clarity, which are very small, i.e., better exposition of why the binary mask is better than the 'soft-mask' methods (Meta-SGD) works.

Originality: As far as I am aware, this work presents a novel method (sparse-MAML) to do the few-shot learning and continual learning.  The code and the experimental details are provided in the supplementary, which would be helpful for reproduction.

Significance: This work seems significant to researchers interested in meta-learning, cross-domain few-shot learning and continual learning. The proposed sparse-MAML is so promising and further testing.

**Time Spent Reviewing:**

12 hours

---

> ### Author Response · Authors · 2021-08-10
> **Reply to Reviewer giJs**
>
> Thank you for the encouraging review and for the suggested improvements. We reply to them point-by-point below:
>
> ---
> > Most of my concerns about clarity, which are very small, i.e., better exposition of why the binary mask is better than the 'soft-mask' methods (Meta-SGD) works.
>
> Meta-learning binary masks instead of step-sizes reinforces the inductive bias towards freezing weights. Our intuition is that determining which weights to update and which to freeze is the essential gradient modulation operation in few-shot and continual learning problems. Accordingly, we believe that the improvements we found over meta-learned step sizes stem from the reduced meta-learning search space. We have expanded the discussion section with this intuition.
>
> ---
> > One small question: Is the sparse-MAML based on transductive batch normalization? It needs to be clarified in the paper.
>
> Yes (for all methods), this was briefly mentioned in the SM but we now included a remark in the main text as well, as it is indeed an important piece of information to interpret the results.

---

### Official Review · Reviewer_C1hv · 2021-07-17

**Rating:** 5
**Confidence:** 3

**Summary:**

The paper investigates how gradient sparsity emerges and adapts in meta-learning and continual learning scenario. The authors suggest to investigate the effects and behavior of sparsity, for example, layer-wise sparsity pattern for few-shot learning, how the sparsity rates adjust to the number of inner-gradient steps, inner-learning rates, and hidden layer size. The authors further perform empirical analysis with continual learning setting, where  combined with La-MAML, the performance improves with proper rate of sparsity.

**Limitations And Societal Impact:**

The authors have adequately addressed the limitations and potential negative societal impact of their work.

**Main Review:**

== Pros ==
The paper involves extensive empirical study, which may be useful for some readers.

== Cons ==
While reading the paper, I really felt that each of the experiments are poorly motivated, without sufficient explanation of why each experiment should appear at a certain point and why each experiment and the corresponding message is important. It seems that the whole paper is simply enumerating many experimental results without proper structures. Taking a few examples,
1. Why is the observation of section 3.1 and Figure 1 important? I see that sparsity levels can be different for different layers and for different initial sparsity ratio, then so what? Such a discussion has not been provided with any proper context, so it is really confusing what we should do next and what will appear next in the paper.
2.  Same for the section 3.2. Personally, this section was a bit informative, such as in Figure 2 and 3, sparsity allows much larger learning rates and longer gradient steps, which is not available for standard MAML. However, I'm still confused why we should investigate this phenomena at this point. What is the context? The paper simply repeats "We next study ..." without properly motivating the readers.
3. Same for the section 3.3. Why do we need to compare with gradient modulation methods at this point? I roughly see that the role of gradient modulation can be overlapping with gradient sparsity, but I'm not sure because there is no such discussion. Furthermore, in L158, why suddenly discuss about stochastic gradient masks? Is the Section 3.3 about stochasticity vs. performance?
4. Same for the section 3.4. I understand that sparsity is beneficial for cross-domain adaptation by looking at the table, but again, what is the context of this experiments? Why the performance improves?

Same for all the other sections. I strongly think that the paper is poorly structured and should be rewritten with clear motivation. I understand that the paper is analysis-styled one, but it should be structured with clear motivation everywhere.

**Time Spent Reviewing:**

6 hours

---

> ### Author Response · Authors · 2021-08-10
> **Reply to Reviewer C1hv**
>
> Thank you for your review. We reply to your specific questions individually below.
>
> ---
> > Why is the observation of section 3.1 and Figure 1 important? I see that sparsity levels can be different for different layers and for different initial sparsity ratio, then so what?
>
> We believe that the results reported in this section are important for at least two reasons. First, they serve to validate that our method can discover sparse learning algorithms. Second, they show that while the patterns of gradient sparsity show rich structure, the level of sparsity is anti-correlated with depth. This is important since there is a long tradition of machine learning algorithms with human-engineered frozen features, with the pattern of feature freezing being determined based on depth (in combination with MAML, see e.g. [31] and [27]). Our method therefore justifies these approaches, while outperforming them, suggesting that it might be preferable to meta-learn which features to freeze. We added this discussion to section 3.1.
>
> ---
> > Same for the section 3.2. Personally, this section was a bit informative, such as in Figure 2 and 3, sparsity allows much larger learning rates and longer gradient steps, which is not available for standard MAML. However, I'm still confused why we should investigate this phenomena at this point. What is the context?
>
> Our hypothesis, which was confirmed by our experiments, is that restricting learning to an appropriate parameter subset allows for longer training and larger changes without overfitting, beyond meta-learning initial parameter values. We now begin section 3.2 by clearly stating this hypothesis. Additionally, the learned sparsity makes MAML more robust, especially to larger inner loop learning rates, which results in performance improvements in all our experiments.
>
> ---
> > Same for the section 3.3. Why do we need to compare with gradient modulation methods at this point? I roughly see that the role of gradient modulation can be overlapping with gradient sparsity, but I'm not sure because there is no such discussion.
>
> As discussed in section 2, sparse-MAML can be understood as a binary gradient modulation method. Second-order methods such as Meta-Curvature [28] modulate gradients by meta-learning pre-conditioning matrices; in meta-SGD [24], these matrices are restricted to be diagonal; sparse-MAML further restricts the diagonal values to be binary. We have added this reminding clarification to section 3.3, paragraph “Sparse-MAML”. Moreover, we have moved Table S1, which shows that meta-learned learning rates also exhibit sparsity, to section 3.3. This result supports our intuition that freezing learning in certain coordinates is the essential gradient modulation operation in few-shot learning.
>
> ---
> > Furthermore, in L158, why suddenly discuss about stochastic gradient masks? Is the Section 3.3 about stochasticity vs. performance?
>
> We agree that the investigation of stochastic gradient masks is better served by a section of its own. We have separated the paragraph from section 3.3 and created a new one. Furthermore, we begin this new section by more clearly stating that our interest in studying stochastic masks is two-fold: as a way to improve meta-optimization with our straight-through estimator; and to determine if stochastic masking is beneficial at meta-test time (which turned out not to be the case in our experiments).
>
> ---
> > Same for the section 3.4. I understand that sparsity is beneficial for cross-domain adaptation by looking at the table, but again, what is the context of this experiments? Why the performance improves?
>
> We wondered whether the patterns of gradient sparsity discovered by our method overfit to the particular task family where they were obtained (miniImageNet classification tasks). This is a natural question to ask as excessive parameter freezing may prevent adaptation to tasks that are too different from those presented during meta-learning. The strong results reported in section 3.4 show that this is not the case, for the datasets considered in the BOIL study. Sparse-MAML is able to outperform both standard MAML and its variants with manually-frozen features, ANIL and BOIL. We now present this motivation in section 3.4.
>
> ---
> We hope that you can take a second look into our paper to more gently appreciate the scientific results contained in it. The gradient based meta-learning literature in few-shot and continual learning has a plethora of methods that either modulate the gradients by a learnable learning rate or have a specific human-engineered split between learnable and fixed parameters. Our paper suggests an alternative, or more precisely an enhancement, to these methods. We show that across many meta-learning domains where gradient modulation is beneficial, gradient sparsity improves it, and helps us interpret the behavior of these algorithms in a more intuitive way. Moreover, we provide a method to meta-learn sparse forms of gradient descent which is efficient and easy to implement.
>
>
> We are more than happy to consider any further suggestions that you may have.

---

> > ### Comment · Reviewer_C1hv · 2021-09-03
> > **Thank you for your clarification and sorry for late reply**
> >
> > Dear authors,
> >
> > I firstly apologize for my late reply. I decided to increase my score to 5, reflecting the significance and importance of the scientific findings done by this paper. However, I could not give acceptance because I think the paper needs a significant amount of rewriting for the reason I mentioned in the first place. Now I think this is a borderline paper, so I will not strongly argue for reject as before.
> >
> > Thank you!

---

> > > ### Author Response · Authors · 2021-09-04
> > > **Thank you for update**
> > >
> > > Dear reviewer C1hv,
> > >
> > > Many thanks for the review update. We will work hard on improving the presentation of our results for the final version of our paper.

---

### Author Response · Authors · 2021-08-10
**Joint reply to all reviewers**

We thank all reviewers for their constructive criticism and useful suggestions that have helped us improve our paper.

Following the reviewers’ comments, we added additional motivation and discussion of the results to the paper and clarified several points, as detailed in our individual replies below. Moreover, we included in the paper a set of class-incremental continual learning experiments on the Split-CIFAR-10 dataset. These new results show that our method consistently outperforms experience replay, a strong baseline in this continual learning setting, while discovering a solution with high levels of parameter freezing in the final output layer.

Given our new results, clarifications and changes to the text we kindly ask the reviewers to reevaluate their scores, if they agree that our paper has improved. We remain available to respond to any further questions the reviewers may have in the upcoming discussion period.

---

> ### Author Response · Authors · 2021-08-20
> **Gradient sparsity improves ResNet-12 few-shot learning performance**
>
> Following reviewer WL2f's question on the scalability of our methods, we ran additional miniImageNet 5-shot learning experiments on a ResNet-12 neural network, which have now concluded.  We summarize our findings in the table below. Our experiments confirm that gradient sparsity emerges on this more complex model accompanied by an increase in test accuracy. We compared vanilla MAML [8] to sparse-MAML, sparse-ReLU-MAML and ANIL [31]; all results were produced using the first-order gradient approximation. We found that this approximation combined with a 25-step-long inner-loop resulted in improved performance over using second-order derivatives and 5-step-long inner-loops (~68% 2nd-order vs. ~72% 1st-order MAML final test accuracy), while being significantly faster.
>
> We found that sparse-ReLU-MAML, which combines gradient sparsity with learning rate modulation, results in the strongest increase in performance over vanilla MAML for this more complex architecture. Our results are competitive with the state-of-the-art for this model (see, e.g., Table 1 from ref. [Baik21]). As for the pattern of gradient sparsity arising in sparse-MAML and sparse-ReLU-MAML, it is qualitatively similar to the one obtained for the smaller convolutional neural network model. Both methods discover solutions with a very high level of parameter freezing in the earlier layers (~90%), which progressively decreases with depth, resulting in an overall level of sparsity of ~30% (note that the final layers of this model have a larger number of parameters).
>
> | Algorithm | Final acc. (%)  | Sparsity (%)
> | :------------- |:-------------:|:-------------:|
> | MAML   | 72.08 +/- 0.55   | -
> | ANIL   | 72.52 +/- 0.38   | -
> | sparse-MAML    | 73.11 +/- 0.45 | 30.37 +/- 1.66
> | sparse-ReLU-MAML | 76.03 +/- 0.15 | 32.24 +/- 1.45
>
> We will include 1-shot learning results in the final version of the paper.
>
> [Baik21] Baik S, Oh J, Hong S, Lee KM (2021). Learning to forget for meta-learning via task-and-layer-wise attenuation, IEEE PAMI 2021, in press. URL: https://ieeexplore.ieee.org/document/9507366

---

### Comment · Reviewer_WL2f · 2021-08-29
**Request to the authors: summary of the updates to appear in the final paper**

Dear authors,

Thank you for all your answers and your active participation in the discussion.

In order to help the reviewers and the AC to take the final decision, I suggest you summarize in a single comment a detailed list of all changes and clarifications that you intend to incorporate into the paper.
This will help us track the differences between the initial and final versions of the paper and verify that these changes are actually happening in case of acceptance. Also, I think this would be particularly helpful for the AC and a good way to evaluate the impact of the review process for this specific paper.

Thanks a lot!

---

> ### Author Response · Authors · 2021-08-30
> **Summary of the updates to appear in the final paper**
>
> Dear reviewers, dear area chair,
>
> Thank you once more for your helpful comments and discussion. To ease your final decision making step, and following reviewer WL2f's suggestion, we provide a detailed summary of changes that will appear in the camera-ready version of our paper:
>
> 1. **ResNet-12 experiments**. We analyzed the miniImageNet 5-shot learning performance of MAML, ANIL, sparse-MAML and sparse-ReLU-MAML on a ResNet-12 model (cf. joint reply update, https://openreview.net/forum?id=8p46f7pYckL&noteId=I5-Y7Kg6gNO). These experiments confirm that the emergence of gradient sparsity and the accompanying increase in generalization are not specific to the convolutional neural network that is commonly studied in visual few-shot learning experiments, and show the scalability of our methods to more complex models. We will additionally include 1-shot learning results in the final version of the paper.
>
> 2. **Stronger experimental results motivation and discussion**. Following reviewers C1hv’s and WL2f’s comments, we will improve the presentation of our experimental results with stronger motivation and expanded discussion. To address reviewer C1hv’s specific points we already made a number of changes to the paper, listed below:
>
>     Added to the end of Section 3.1:
>
>        These findings validate that our method can discover sparse learning algorithms. Moreover, they show that the level of sparsity is anti-correlated with depth. This is important since there is a long tradition of machine learning algorithms with human-engineered frozen features, with the pattern of feature freezing being determined based on depth (in combination with MAML, see e.g. [31] and [27]). Our method therefore justifies these approaches, while outperforming them, suggesting that it might be preferable to meta-learn which features to freeze. Our findings hold when applying our method to a deeper and wider residual neural network (ResNet-12) model, where we observe the same trend of decreasing gradient sparsity with depth emerge.
>
>     Added to the beginning of Section 3.2:
>
>        We hypothesize that restricting learning to an appropriate parameter subset allows for longer training and larger changes without overfitting, beyond meta-learning initial parameter values. To verify this hypothesis we scan over different inner-loop learning rates and lengths and compare the resulting test set performances of MAML and sparse-MAML.
>
>     Changed the beginning of Section 3.3 to:
>
>        Sparse-MAML can be understood as a binary gradient modulation method. Second-order methods such as Meta-Curvature [28] modulate gradients by meta-learning pre-conditioning matrices; in meta-SGD [24], these matrices are restricted to be diagonal; sparse-MAML further restricts the diagonal values to be binary. From this point of view, sparse-MAML is the least expressive form of gradient modulation. Surprisingly, we find that despite its reduced expressiveness, sparse-MAML recovers the performance improvements achieved by the more sophisticated alternatives, significantly improving the performance of standard MAML (cf Table 1).
>
>     Moved the detailed description of sparse-MAML+ from the SM (page 2) to a new main text subsection dedicated to it; added the following passage motivating the study of sparse-MAML+:
>
>        Our interest in studying stochastic masks is two-fold: as a way to improve meta-optimization with our straight-through estimator; and to determine if stochastic masking is beneficial at meta-test time.
>
>     Added to the beginning of section 3.4:
>
>        We now investigate whether the patterns of gradient sparsity discovered by our method overfit to the particular task family where they were obtained, namely, few-shot miniImageNet classification tasks. This is an important question, since excessive parameter freezing may prevent adaptation to tasks that are too different from those presented during meta-learning.
>
> 3. **Expanded discussion**. Per reviewer giJs's question, we added the following intuition to the Discussion section:
>
>        Meta-learning binary gradient masks instead of step-sizes reinforces the inductive bias towards freezing weights. Our intuition is that determining which weights to update and which to freeze is the essential gradient modulation operation in few-shot and continual learning problems. This intuition is confirmed by the strong performance of sparse-MAML, which isolates the improvements brought by freezing parameters from those that stem from finer forms of gradient modulation. The effectiveness of sparse gradients is further supported by the large fraction of frozen parameters found when meta-learning rectified learning rates using our straight-through estimation method.
>
> 4. **On transductive BatchNorm layers**. Following the questions of reviewers giJs and WL2f, we added to the main text a clarifying note stating that all main text experiments use transductive batch normalization. Furthermore, we will report non-transductive miniImageNet few-shot learning results in the SM; we are currently experimenting with TaskNorm (Bronskill et al., ICML2020) to avoid the standard MAML handling of batch normalization layers.
>
> 5. **Class-incremental Split-CIFAR-10 results**. We complemented our continual learning investigation of gradient sparsity in La-MAML with new results on a streaming Split-CIFAR-10 class-incremental learning problem (cf. post https://openreview.net/forum?id=8p46f7pYckL&noteId=PFeJO8RGUdc). Our results confirm that La-MAML's improvements over experience replay (a relevant baseline in class-incremental learning) translate to this challenging setting, and reveal that meta-learning finds a solution with large gradient sparsity in the final output layer. GEM, MER, and ReLU La-MAML performances will be added to the final version of the paper.
>
> 6. **Clarification on BTI metric.** Per reviewer ah52's question on the BTI metric, we added the following remark to section 4:
>
>        The lower baseline BTI (smaller average change in absolute accuracy) can be explained by the lower overall accuracies achieved by it.
>
> 7. **Frozen bias parameters.** Reviewer ah52's asked whether the consistent freezing of bias parameters by our methods means that they are useful task-shared parameters, or unneeded altogether. We will report in the SM the results of a small ablation experiment investigating the performance of a CNN without biases.
>
> 8. **Discussion on gradient regularization.** Following reviewer WL2f's comments on gradient regularization, we added to following point to the Discussion section:
>
>        We found that gradient sparsity emerges in meta-learning when optimizing for generalization on a heldout dataset, without any explicit regularization towards it. Compared to gradient regularization methods, an important practical advantage of our approach is that it does not require evaluating second derivatives. This advantage allowed us to apply our methods to a ResNet model. Comparing the performance of sparse-MAML and sparse-ReLU-MAML to more expensive second-order methods, including those involving gradient regularization, is an interesting direction for future work.
>
> 9. **On meta-shrinkage**. Reviewer WL2f asked for a comparison to meta-shrinkage [6]. We added the results from Table 10 of [6] to our Table 1 and expanded our discussion of this method in section 3.4, highlighting the differences in approach: we use first-order straight-through gradient mask learning, while meta-shrinkage learns a parameter-specific L2-regularizer using implicit differentiation. We now discuss that while meta-shrinkage also discovers a similar pattern of frozen task-shared features, on miniImageNet it does not improve learning performance as much as sparse-MAML.
>
> 10. **Gradient sparsity in sparse-ReLU-MAML**. We now present Table S1 (which reports a large fraction of frozen parameters in sparse-ReLU-MAML) in section 3.4 of the main text to highlight that gradient sparsity emerges also when meta-learning rectified learning rates.

---

### Author Response · Authors · 2021-08-31
**Happy to provide final clarifications**

Dear reviewers,

Thank you once again for your useful comments. The end of the rolling discussion period is coming close, and we see that some of you have not yet updated your reviews in response to our new results and proposed changes addressing your concerns. In case something is unclear, please do not hesitate in contacting us before the interactive discussion period ends.

---

### Decision · Program_Chairs · 2021-09-28

**Decision:**

Accept (Poster)

**Comment:**

Echoing the reviewers, I think this work is of interest to the community, and I think the experimental section is thoroughly done. Taking into account the engagement from the authors, and the correction they proposed and showed in this comments (which I hope will be reflected in the camera ready) I think the paper matches the requirements for being accepted at the conference.

**Consistency Experiment:**

NeurIPS has a long history of experimentation. In 2014, NeurIPS ran an experiment in which 10% of submissions were reviewed by two independent committees to quantify the randomness in the review process. This year, we repeated a variant of this experiment to see how the quality of the review process has changed over time.  This paper was part of the experiment and was therefore assigned to two committees (consisting of reviewers, an Area Chair, and a Senior Area Chair) that reached independent decisions.  If both committees made the same recommendation, this recommendation was followed. If a single committee recommended acceptance, the paper was accepted (with the exception of a few cases in which the other committee identified what we considered a fatal flaw, e.g., an error in a key result).

Both committees reached the same decision: **Accept (Poster)**

The other committee assigned to the paper recommended **Accept (Poster)**.  You can find the other set of reviews, along with any follow up discussion with the authors here:
https://openreview.net/forum?id=CxefshFHEqh